# Approximation of Log-Partition Function in Policy Mirror Descent Induces Implicit Regularization for LLM Post-Training

**Zhenghao Xu** [1] [*]  **Qin Lu** [2]  **Changlong Yu** [2]  **Tuo Zhao** [1]

## Abstract

Policy mirror descent (PMD) provides a principled framework for reinforcement learning (RL) by iteratively solving KL-regularized policy improvement subproblems. While this approach has been adopted in training advanced LLMs such as Kimi K1.5/K2, the ideal closed-form PMD updates require reliable partition function estimation, a significant challenge when working with limited rollouts in the vast action spaces of LLMs. We investigate a practical algorithm, termed PMD-MEAN, that approximates the log-partition term with the mean reward under the sampling policy and performs regression in log-policy space. Specifically, we characterize the population solution of PMD-MEAN and demonstrate that it implicitly optimizes mirror descent subproblems with an adaptive mixed KL–$\chi^2$ regularizer. This additional $\chi^2$ regularization constrains large probability changes, producing more conservative updates when expected rewards are low and enhancing robustness against finite-sample estimation errors. Experiments on math reasoning tasks show that PMD-MEAN improves stability and remains competitive in the stale-rollout settings while enabling higher throughput through larger rollout batches. These findings deepen our understanding of PMD-MEAN and illuminate pathways toward principled improvements in RL algorithms for LLMs. Code is available at https://github.com/horizon-llm/OpenKimi.

## 1. Introduction

Reinforcement learning (RL) has become a standard paradigm for enhancing post-training of large language models (LLMs) on reasoning tasks and agentic objectives (OpenAI, 2024; Guo et al., 2025). Despite diverse implementation approaches, most RL algorithms can be formalized as *regularized policy improvement*, an iterative method that updates policies to maximize rewards while maintaining proximity to reference policies.

Policy mirror descent (PMD, Geist et al. 2019; Tomar et al. 2022) provides a canonical formalization of this approach by iteratively solving KL-regularized improvement subproblems. In theory, these subproblems admit elegant closed-form solutions that reweight the current policy and renormalize using the partition function. In practice, however, reliably estimating this partition function and fitting the ideal target from finite rollouts presents significant challenges, particularly in the large action space of LLM post-training.

A common approach to solving KL-regularized subproblems involves applying policy gradient methods (Williams, 1992) directly to the regularized objective, either by incorporating regularization into the reward function or adding an explicit KL penalty. Methods such as TRPO (Schulman et al., 2015), PPO (Schulman et al., 2017), RLOO (Ahmadian et al., 2024), and GRPO (Shao et al., 2024) use rollout samples to construct surrogate losses and perform optimally when rollouts are from the current policy, i.e., on-policy.

However, modern efficient RL implementations increasingly leverage large generation batches or asynchronous rollouts to prevent computational bottlenecks from long-tail generations (Noukhovitch et al., 2024; Fu et al., 2025). These approaches typically incur a *staleness tax*: the sampling policy predates the policy being updated, creating a fundamental training/inference mismatch. This mismatch introduces instability that practitioners attempt to mitigate through importance weighting with clipping or similar heuristics (Yao et al., 2025; Liu et al., 2025). While partially effective, these remedial techniques substantially complicate both implementation and theoretical analysis.

This paper investigates an alternative minimalist approach popularized by Kimi K1.5/K2 (Team et al., 2025b;a) that fundamentally reframes the problem. Rather than attempting to mitigate off-policy-ness through complex correction mechanisms and heuristics, this algorithm adopts an off-

---

[*]Work done during internship at Amazon. [1]Georgia Institute of Technology [2]Amazon. Correspondence to: Zhenghao Xu <zhenghaoxu@gatech.edu>, Tuo Zhao <tourzhao@gatech.edu>.

*Proceedings of the 43rd International Conference on Machine Learning*, Seoul, South Korea. PMLR 306, 2026. Copyright 2026 by the author(s).

policy regression perspective on PMD. Specifically, instead of fitting the exact partition-normalized target, the method approximates the log-partition term with the *mean reward* under the sampling policy and fits a regression target directly in log-policy space. We refer to this practical algorithm as PMD-MEAN (or "Kimi-style PMD").

While this mean-reward approximation remains accurate under strong regularization conditions, it can diverge significantly from the partition-normalized update when using smaller regularization, typical in practice. This divergence raises a fundamental question:

*What does PMD-MEAN optimize exactly, and what are the algorithmic consequences of that objective?*

**Our results.** We address this fundamental question by deriving a closed-form characterization of the PMD-MEAN population solution. While the ideal KL-regularized PMD update produces a standard Boltzmann reweighting, our analysis reveals that PMD-MEAN generates an update involving the Lambert-$W$ function.

Furthermore, we show that this update is mathematically equivalent to performing mirror descent with a *mixed* KL–$\chi^2$ regularizer, where the $\chi^2$ weight *adapts* dynamically based on the mean reward under the current policy. This additional $\chi^2$ term imposes stronger penalties on probability changes compared to KL alone, and the effect is particularly pronounced when the mean reward is low. This adaptive regularization effectively moderates the convergence rate during the early phases of training, providing a principled explanation for the algorithm's empirical stability.

Our further analysis demonstrates that, compared to fitting the partition-normalized target (PMD-PART), PMD-MEAN exhibits significantly reduced sensitivity to finite-sample errors when rollouts are limited. This characteristic substantially decreases the risk of overfitting to misestimated targets. The finding provides a theoretical explanation for PMD-MEAN's enhanced stability in practical applications: The implicitly induced $\chi^2$ regularization introduces additional robustness that is valuable in the data-constrained scenarios typical of LLM post-training.

**Contributions.** We make the following contributions:

• **Exact characterization of PMD-MEAN.** We derive the closed-form PMD-MEAN solution in policy space and establish its equivalence to a mirror-descent subproblem with an adaptive mixed KL–$\chi^2$ regularizer.

• **Regularization and stability mechanism.** We demonstrate that the induced $\chi^2$ term provides direct control over probability ratios, offering substantial regularization even when the nominal KL coefficient is minimal.

• **Convergence analysis.** Under standard assumptions, we develop an inexact-PMD style convergence analysis that distinguishes PMD-MEAN from PMD-PART and precisely characterizes their fundamental differences.

• **Experimental validation.** Through experiments on math reasoning tasks, we empirically validate the predicted stability behavior and observe gains over stale-rollout GRPO in our evaluated runs.

## 2. Preliminaries

For simplicity, we formulate RL for LLM post-training as a contextual bandit problem. Let $x \in \mathcal{X}$ be an input prompt (state) and $y \in \mathcal{Y}$ represent a generated response (action) of finite length, with $r(x, y) \in [0, 1]$ denoting a bounded reward function. A language model policy $\pi \colon \mathcal{X} \to \Delta(\mathcal{Y})$, which maps states to distributions over actions, induces the expected reward:

$$J(\pi) := \mathbb{E}_{x \sim \mathcal{D}} \mathbb{E}_{y \sim \pi(\cdot|x)} \big[ r(x, y) \big],$$

where $\mathcal{D}$ is the distribution over prompts (e.g., a dataset). In practice, LLM policies are implemented as compositions of token-wise softmax distributions, which assign non-zero probability to every token in the vocabulary. Consequently, we can reasonably assume that $\pi$ has full support over the entire action space of possible responses, that is, $\pi(y \mid x) > 0$ for all $x \in \mathcal{X}$ and $y \in \mathcal{Y}$.

This contextual bandit abstraction treats a full response as one action, matching sequence-level reinforcement learning with verifiable reward (RLVR) training. It does not model (stochastic) multi-turn state transitions or token-level credit assignment. These full-MDP issues require additional control of value estimation and state-distribution shift and are beyond the present scope.

**Policy mirror descent.** At global step $t$, KL-regularized policy mirror descent (PMD; Geist et al. 2019; Tomar et al. 2022) updates policy $\pi_t$ by solving the following optimization problem for each state $x$:

$$\pi_{t+1}(\cdot \mid x) = \operatorname*{argmax}_{\pi(\cdot|x) \in \Delta(\mathcal{Y})} \mathbb{E}_{y \sim \pi(\cdot|x)}[r(x, y)]$$
$$- \tau \cdot \mathrm{KL}\left(\pi(\cdot \mid x) \,\|\, \pi_t(\cdot \mid x)\right), \quad (1)$$

where $\tau > 0$ is the regularization parameter that controls the strength of regularization. The KL divergence between distributions $p$ and $q$ over $\mathcal{Y}$ is $\mathrm{KL}(p \,\|\, q) := \mathbb{E}_{y \sim p}\left[\log \frac{p(y)}{q(y)}\right]$.

This KL-regularized optimization problem in (1) admits the following unique closed-form solution:

$$\pi_{t+1}(y \mid x) = \frac{\pi_t(y \mid x) \exp(r(x, y)/\tau)}{Z_t(x)}, \quad (2)$$

where $Z_t(x) := \mathbb{E}_{y \sim \pi_t(\cdot|x)}\left[e^{r(x,y)/\tau}\right]$ is the partition function that ensures proper normalization. This update resem-

bles a Boltzmann distribution that exponentially reweights the previous policy according to the reward signal.

**Fitting the target by regression.** The ideal update in (2) is computationally infeasible in large action spaces as it requires partition function evaluation and per-action probability assignment. A direct off-policy approach to approximate this update is to fit the target in log-policy space using squared regression. After collecting rollouts $y \sim \pi_t(\cdot \mid x)$ and evaluating rewards $r(x, y)$, we define the state-dependent target log-ratio:

$$s^\star_{\text{part}}(x, y) := \log \frac{\pi_{t+1}(y \mid x)}{\pi_t(y \mid x)} = \frac{r(x, y)}{\tau} - \log Z_t(x),$$

where $\pi_{t+1}$ is defined according to Equation (2). This leads to the following squared regression loss:

$$\mathcal{L}_{\text{part}}(\pi) :=$$
$$\mathbb{E}_{x \sim \mathcal{D}} \mathbb{E}_{y \sim \pi_t(\cdot \mid x)} \left[ \frac{1}{2} \left( \log \frac{\pi(y \mid x)}{\pi_t(y \mid x)} - s^\star_{\text{part}}(x, y) \right)^2 \right].$$
$$(3)$$

If the policy class can represent the target distribution exactly (i.e., is realizable) and $Z_t(x)$ is known precisely, then minimizing (3) recovers the optimal policy update. However, in practice, $Z_t(x)$ must be estimated from the same finite set of rollouts, which introduces significant estimation error. For large action spaces and small regularization parameters $\tau$, this estimation can be highly unstable, leading to pathological update behavior (see Section 5). We refer to this direct fitting approach as PMD-PART.

**PMD-MEAN: Log-partition approximation.** Instead of fitting $s^\star_{\text{part}}$, Team et al. (2025b) propose an alternative approach that approximates the log-partition function with the average reward, resulting in a modified loss function. Specifically, we define the advantage function $\Delta(x, y)$ under $\pi_t$ with mean reward baseline:

$$\Delta(x, y) := r(x, y) - \mathbb{E}_{y' \sim \pi_t(\cdot \mid x)}[r(x, y')].$$

With this definition, the regression objective becomes:

$$\mathcal{L}_{\text{mean}}(\pi) :=$$
$$\mathbb{E}_{x \sim \mathcal{D}} \mathbb{E}_{y \sim \pi_t(\cdot \mid x)} \left[ \frac{1}{2} \left( \log \frac{\pi(y \mid x)}{\pi_t(y \mid x)} - \frac{\Delta(x, y)}{\tau} \right)^2 \right]. \quad (4)$$

We refer to this practical variant as PMD-MEAN, which has been adopted to train advanced models such as Kimi K1.5 and K2 (Team et al., 2025b;a). In practice, the expected reward $\mathbb{E}_{y' \sim \pi_t(\cdot \mid x)}[r(x, y')]$ can be efficiently estimated using a per-prompt Monte Carlo average over sampled responses.

## 3. Implicit Regularization of PMD-MEAN

The approximation adopted by PMD-MEAN is accurate when $\tau \to \infty$. However, a large $\tau$ will significantly slow down convergence, and the average reward deviates significantly from the log-partition function when $\tau$ becomes small (Figure 1, left). Therefore, the solution of (4) may differ significantly from the ideal target $\pi_{t+1}$ in (2), and thus no longer corresponds to the solution of the KL subproblem (1), even with infinite samples provided (Figure 1, right).

### 3.1. Exact Solution of PMD-MEAN

To understand the actual objective of PMD-MEAN, we characterize the population minimizer of (4). For simplicity, we omit $x$ and consider a single state, writing $\pi_t(y)$ and $\Delta_y$.

**Theorem 3.1** (PMD-MEAN solution). *Assume $\pi_t(y) > 0$ for all $y \in \mathcal{Y}$. Let $\Delta_y := r(y) - \mathbb{E}_{y' \sim \pi_t}[r(y')]$ denote the mean-baseline advantage. Then the unique minimizer of (4) over the probability simplex satisfies*

$$\pi_{t+1}(y) = \pi_t(y) \exp \left( \frac{\Delta_y}{\tau} - W \left( \frac{\lambda}{\tau^2} \exp \left( \frac{\Delta_y}{\tau} \right) \right) \right), \quad (5)$$

*where $W(\cdot)$ is the principal branch of the Lambert-W function (inverse of $f(w) = w \cdot e^w$) and $\lambda \geq 0$ is a normalization constant chosen such that $\sum_y \pi_{t+1}(y) = 1$. Moreover, defining $A := \mathbb{E}_{\pi_t}[\exp(\Delta_y/\tau)]$ and $B := \mathbb{E}_{\pi_t}[\exp(2\Delta_y/\tau)]$, the normalization constant satisfies*

$$\tau^2 \frac{A(A-1)}{B} \leq \lambda \leq \tau^2 \log A. \quad (6)$$

*For binary rewards $r \in \{0, 1\}$, the non-degenerate case $p = \mathbb{E}_{\pi_t}[r(y)] \in (0, 1)$ satisfies*

$$\lambda = \begin{cases} \frac{1}{2} p(1-p) + O(\tau^{-1}), & \tau \to \infty, \\ \tau p(1-p)(1 + o(1)), & \tau \to 0. \end{cases} \quad (7)$$

By Theorem 3.1, the solution of PMD-MEAN has its action probabilities heterogeneously normalized by the Lambert-$W$ function, as opposed to the KL solution (2) where the normalization term is the log-partition function that is independent of the action $y$. Given the monotonicity of $W$, actions with larger $\Delta_y$ will get their probability suppressed compared to the KL solution, while actions with smaller $\Delta_y$ will not be punished as hard. Therefore, PMD-MEAN update is less aggressive than PMD-PART.

More precisely, suppose the reward is binary and the average reward under $\pi_t$ is $p$. We consider the ratio $\pi_{t+1}/\pi_t$ on positive and negative actions when $\tau$ is small. For positive actions, $r(y) = 1$, PMD-MEAN yields

$$\frac{\pi_{t+1}^{\text{mean}}(y)}{\pi_t(y)} = \frac{1}{p} - \frac{1-p}{p} e^{-p/\tau} (1 + o(1)), \quad (8)$$

while PMD-PART yields

$$\frac{\pi_{t+1}^{\text{part}}(y)}{\pi_t(y)} = \frac{1}{p} - \frac{1-p}{p^2} e^{-1/\tau} + O(e^{-2/\tau}). \quad (9)$$

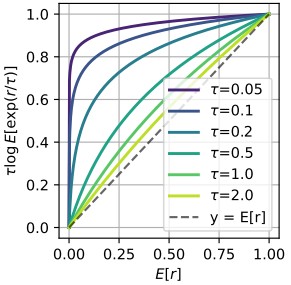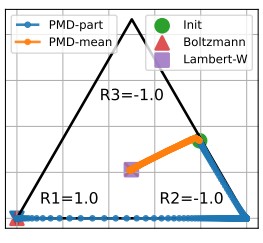

*Figure 1.* Left: Scaled log-partition function vs average reward assuming binary rewards. The gap is significant for moderate $\tau$. Right: Illustration of PMD-MEAN and PMD-PART converging to different subproblem solutions in the probability simplex.

While both ratios approach $1/p$ from below when $\tau \to 0$, the gap is much larger in PMD-MEAN when $p$ is small (e.g., early phase of training), hence giving a more conservative update. For negative actions, the separation is clearer: For $r(y) = 0$, PMD-MEAN yields

$$\frac{\pi_{t+1}^{\text{mean}}(y)}{\pi_t(y)} = e^{-p/\tau}(1 + o(1)), \qquad (10)$$

while PMD-PART yields

$$\frac{\pi_{t+1}^{\text{part}}(y)}{\pi_t(y)} = \frac{1}{p}e^{-1/\tau} + O(e^{-2/\tau}). \qquad (11)$$

When $p$ is small, the difference is significant, as illustrated in Figure 2. Full derivations are provided in Section B.2.

*Remark* 3.2 (Length normalization). The main theory analyzes the sequence-level objective in (4). Our experiments use the length-normalized implementation in Section A. Section B.5 derives the analogous Lambert-$W$ form, where normalization induces an action-dependent quadratic ratio penalty.

### 3.2. PMD-MEAN as Mirror Descent with Mixed KL–$\chi^2$ Regularization

The Lambert-$W$ closed form is useful for analysis, while a more transparent insight is that PMD-MEAN is *exactly* solving a different regularized policy improvement problem. Let $\chi^2(p \parallel q) := \mathbb{E}_{y \sim q}\left[\left(\frac{p(y)}{q(y)} - 1\right)^2\right]$ denote the $\chi^2$-divergence between two distributions $p$ and $q$ over $\mathcal{Y}$. The following proposition shows that the PMD-MEAN update is equivalent to mirror descent with an additional $\chi^2$ penalty.

**Proposition 3.3** (Equivalent mixed KL–$\chi^2$ subproblem). *Fix a state $x$ (omitted for brevity). Let $\pi_{t+1}$ be the PMD-MEAN population solution in Theorem 3.1 and $\lambda$ be the same normalization constant. Then $\pi_{t+1}$ is the solution to*

$$\pi_{t+1} = \underset{\pi \in \Delta(\mathcal{Y})}{\operatorname{argmax}} \, \mathbb{E}_{y \sim \pi}\big[r(y)\big]$$

$$- \tau \text{KL}(\pi \parallel \pi_t) - \frac{\lambda}{2\tau}\chi^2(\pi \parallel \pi_t). \quad (12)$$

Proposition 3.3 follows directly from writing out the KKT conditions of both problems (see Section B.3).

The $\chi^2$ penalty directly suppresses large policy ratio spikes and provides a stronger regularization compared to KL, as $\text{KL}(p \parallel q) \le \chi^2(p \parallel q)$ for $p, q$ with full support. Moreover, Theorem 3.1 shows the effective strength $\lambda/\tau$ is *adaptive*. For binary rewards, $\lambda/\tau$ remains $O(1)$ even as $\tau \to 0$, which implies PMD-MEAN still regularizes updates when the nominal KL regularization is small.

*Remark* 3.4 (Connection to $\chi^2$ preference optimization). The mixed KL–$\chi^2$ regularizer has appeared in $\chi^2$-regularized preference optimization ($\chi$PO, Huang et al. 2024), which provides provable guarantees to avoid overoptimization in KL-regularized preference optimization. Our results show that PMD-MEAN can be interpreted as an online version of this idea, with an adaptive coefficient $\lambda$ tied to the rollout reward distribution.

*Remark* 3.5 (Policy ratios compared to Huang et al. 2024). Huang et al. (2024) show in their Proposition 4.2 that mixed KL–$\chi^2$ regularized problem yields (in our notations)

$$\exp\left(-\frac{R_{\max}}{\tau}\right) \lesssim \frac{\pi_{t+1}^{\text{mix}}}{\pi_t} \lesssim 1 + \frac{R_{\max}}{\tau}, \qquad (13)$$

while the KL-regularized problem yields

$$\exp\left(-\frac{R_{\max}}{\tau}\right) \lesssim \frac{\pi_{t+1}^{\text{KL}}}{\pi_t} \lesssim \exp\left(\frac{R_{\max}}{\tau}\right), \qquad (14)$$

where $r \in [0, R_{\max}]$ is the bound of rewards. This highlights a worst-case polynomial vs. exponential contrast in $1/\tau$ for the upper policy ratio control. However, these bounds are uniform in $R_{\max}$ and can be vacuous in regimes relevant to our binary reward analysis. Particularly, (14) neglects the partition normalizer that also grows with $1/\tau$. In contrast, our setting admits substantially sharper and distribution-dependent behavior for small $\tau$. As shown in (8) and (9), both ratios share the same constant upper bound $1/p$, with different exponential rates of approaching this upper bound. Therefore, the gap between mixed and KL divergence in our setting is more of a distinction in the distribution-dependent exponential rates, i.e., $O(e^{-p/\tau})$ vs. $O(e^{-1/\tau})$, instead of polynomial vs. exponential.

*Remark* 3.6 (Reward scope). The Lambert-$W$ characterization and mixed KL–$\chi^2$ reformulation do not rely on binary rewards. We use the binary model only when deriving sharp pass-rate-dependent constants and target-estimation comparisons. For general bounded rewards, the same normalization-induced ratio-control mechanism applies, but the resulting bounds are distribution-dependent rather than closed-form in a single pass rate.

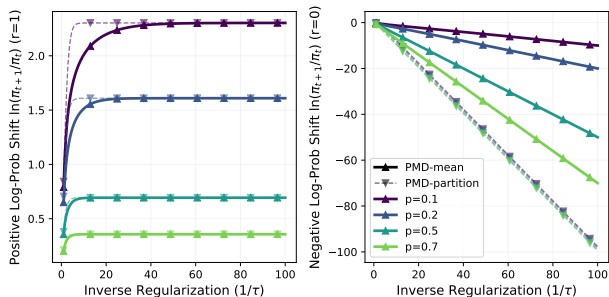

*Figure 2.* The (log) probability ratio of updates in PMD-MEAN is more conservative than that in PMD-PART for binary rewards.

# 4. Implications on Convergence

We present an inexact-PMD convergence analysis from the regression view to illustrate the implications of the implicit regularization on convergence. For clarity, we still suppress $x$ and analyze one state. The extension to averaging over the prompt distribution $x \sim \mathcal{D}$ is standard.

## 4.1. One-Step Policy Improvement

Recall $J(\pi) \coloneqq \mathbb{E}_{y \sim \pi}[r(y)]$ where $r(y) \in [0, 1]$. At iteration $t$, let $\pi_{t+1}^\star$ denote the ideal target update in PMD-PART or PMD-MEAN, and define its target log-ratio $s^\star(y) \coloneqq \log \frac{\pi_{t+1}^\star(y)}{\pi_t(y)}$. Let $\Pi$ be the policy class and define, for each $\pi \in \Pi$, $s_\pi(y) \coloneqq \log \frac{\pi(y)}{\pi_t(y)}$. The goal of PMD-PART and PMD-MEAN is to fit the ideal target $s^\star$ with $s_\pi$ by minimizing the population loss

$$\mathcal{L}_t(\pi) \coloneqq \frac{1}{2} \mathbb{E}_{y \sim \pi_t}\big[(s_\pi(y) - s^\star(y))^2\big]. \qquad (15)$$

In practice, $\pi_{t+1}^\star$ depends on population quantities (e.g., $\mathbb{E}_{\pi_t}[r]$ or $\log Z_t$) that are not exactly available, so one instead forms an estimated target update $\widetilde{\pi}_{t+1}^\star$ via finite MC samples. Given i.i.d. samples $y_1, \ldots, y_n \sim \pi_t$, we minimize the empirical loss

$$\widehat{\mathcal{L}}_t(\pi) \coloneqq \frac{1}{2n} \sum_{i=1}^{n} (s_\pi(y_i) - \widetilde{s}_{-i}^\star(y_i))^2, \qquad (16)$$

where the target at $y_i$ becomes the leave-one-out (LOO) estimated version $\widetilde{s}_{-i}^\star(y_i)$. For PMD-MEAN,

$$\widetilde{s}_{-i}^\star(y_i) = \frac{1}{\tau}\Big(r(y_i) - \frac{1}{n-1} \sum_{j \neq i} r(y_j)\Big). \qquad (17)$$

For PMD-PART,

$$\widetilde{s}_{-i}^\star(y_i) = \frac{r(y_i)}{\tau} - \log\Big(\frac{1}{n-1} \sum_{j \neq i} e^{r(y_j)/\tau}\Big). \qquad (18)$$

Let $\widehat{\pi}_{t+1}$ be an approximate ERM solution and set $\pi_{t+1} \coloneqq \widehat{\pi}_{t+1}$. We make the following assumptions for analysis.

**Assumption 4.1** (Realizability). $\pi_{t+1}^\star \in \Pi$.

**Assumption 4.2** (Bounded optimization error). $\widehat{\mathcal{L}}_t(\widehat{\pi}_{t+1}) \leq \inf_{\pi \in \Pi} \widehat{\mathcal{L}}_t(\pi) + \epsilon_{\mathrm{opt}}$.

**Assumption 4.3** (Bounded log-ratio). There exist $B, B_+ > 0$ such that for all $\pi \in \Pi$ and $y \in \mathcal{Y}$,

$$s_\pi(y) \leq B_+, \quad |s_\pi(y)| \leq B. \qquad (19)$$

**Assumption 4.4** (Finite policy class). $|\Pi| < \infty$.

Assumption 4.1 is standard in the sample-efficient RL literature (Foster & Rakhlin, 2023). Assumption 4.2 is a generic assumption that allows focusing on statistical rates without involving overcomplicated subproblem optimization dynamics. Assumption 4.3 can be achieved by restricting the policy class via clipping. Assumption 4.4 is for simplicity, and one can extend it to general policy classes with $|\Pi|$ replaced by other complexity metrics, e.g., covering number. Under these assumptions, we have the following lemma that characterizes the error of the approximate ERM solution $\widehat{\pi}_{t+1}$.

**Lemma 4.5** (Empirical minimization). *For global step $t$, suppose $y_1, \ldots, y_n \sim \pi_t$ are i.i.d. samples. Let $\mathcal{L}_t(\pi)$ and $\widehat{\mathcal{L}}_t(\pi)$ denote the population and empirical losses defined in (15) and (16), respectively. Define the target mismatch*

$$\Delta_i \coloneqq \widetilde{s}_{-i}^\star(y_i) - s^\star(y_i), \quad \overline{\Delta^2} \coloneqq \frac{1}{n} \sum_{i=1}^{n} \Delta_i^2.$$

*Under Assumptions 4.1 to 4.4, for any $\delta \in (0, 1)$, with probability at least $1 - \delta$,*

$$\mathcal{L}_t(\widehat{\pi}_{t+1}) \lesssim \frac{B^2 \log(|\Pi|/\delta)}{n} + \epsilon_{\mathrm{opt}} + \overline{\Delta^2}. \qquad (20)$$

The proof is provided in Section C.1. The ERM solution error is then translated into the gap between $J(\pi_{t+1}^\star)$ and $J(\widehat{\pi}_{t+1})$ that affects the one-step policy improvement.

**Theorem 4.6** (One-step policy improvement). *Under Assumptions 4.1 to 4.4, suppose the ideal population update $\pi_{t+1}^\star$ satisfies, for some $\eta_t \in (0, 1]$,*

$$1 - J(\pi_{t+1}^\star) \leq (1 - \eta_t)\big(1 - J(\pi_t)\big). \qquad (21)$$

*Let $\pi_{t+1} \coloneqq \widehat{\pi}_{t+1}$ and $\overline{\Delta^2}$ be as in Lemma 4.5. Then for $\delta \in (0, 1)$, with probability at least $1 - \delta$,*

$$
\begin{aligned}
&1 - J(\pi_{t+1}) \\
&\leq (1 - \eta_t)\big(1 - J(\pi_t)\big) \\
&\quad + O\Big(e^{B_+/2}\Big(B\sqrt{\frac{\log(|\Pi|/\delta)}{n}} + \sqrt{\epsilon_{\mathrm{opt}}} + \sqrt{\overline{\Delta^2}}\Big)\Big). \qquad (22)
\end{aligned}
$$

## 4.2. Instantiation and Separation

We now specialize Theorem 4.6 to the binary reward model $r(y) \in \{0, 1\}$ with $p_t := J(\pi_t) \in (0, 1)$ in the small $\tau > 0$ regime, and instantiate (i) the ideal improvement rate $\eta_t$ in (21), (ii) the bounded log-ratio constants $(B, B_+)$ in Assumption 4.3, and (iii) the target estimation error $\overline{\Delta}^2$. These quantities highlight a separation between PMD-PART and PMD-MEAN at the early phase of training where $p_t$ is small: PMD-PART has a faster *ideal* convergence rate, while PMD-MEAN can be more *robust* to statistical errors under finite rollouts.

### 4.2.1. IDEAL CONVERGENCE RATE $\eta_t$

The ideal improvement rate is a direct consequence of our analysis in Section 3, and reveals the behavior when the rollout sample size $n$ is large.

**Proposition 4.7** (Ideal contraction for PMD-MEAN with small $\tau$). *Let $\pi_{t+1}^\star$ be the ideal update (5) with binary rewards. Then as $\tau \to 0$ we have*

$$\eta_t^{\mathrm{mean}} = 1 - \exp\left(-\frac{p_t}{\tau}\right)(1 + o(1)). \tag{23}$$

**Proposition 4.8** (Ideal contraction for PMD-PART). *Let $\pi_{t+1}^\star$ be the ideal update (2) with binary rewards. Then (21) holds with*

$$\eta_t^{\mathrm{part}} = 1 - \frac{1}{1 - p_t + p_t e^{1/\tau}}. \tag{24}$$

The proofs are provided in Sections C.3 and C.4. The ideal convergence rates in PMD-PART and PMD-MEAN both approach 1 when $\tau \to 0$, leading to one-step convergence. Meanwhile, PMD-PART approaches this rate faster than PMD-MEAN when $p_t < 1$ is small.

### 4.2.2. LOG-RATIO BOUNDS $(B, B_+)$ COMPATIBLE WITH REALIZABILITY

In Theorem 4.6, the error of the inexact update depends on $B$ and $B_+$. We compute the smallest $(B, B_+)$ compatible with the realizability of the ideal target in the binary model.

**Proposition 4.9** (Log-ratios for PMD-MEAN with small $\tau$). *Consider binary rewards and the PMD-MEAN target $s^\star(y) = \log \frac{\pi_{t+1}^{\mathrm{mean}}(y)}{\pi_t(y)}$. Then for fixed $p_t \in (0, 1)$ and $\tau \to 0$, the log-ratio bounds are*

$$\exp(B_{+,t}^{\mathrm{mean}}) = \frac{1}{p_t} - \frac{1 - p_t}{p_t}e^{-p_t/\tau}(1 + o(1)),$$

$$B_t^{\mathrm{mean}} = \frac{p_t}{\tau} + o(1).$$

**Proposition 4.10** (Log-ratios for PMD-PART with small $\tau$). *Consider binary rewards and the PMD-PART target*

$s^\star(y) = \log \frac{\pi_{t+1}^{\mathrm{part}}(y)}{\pi_t(y)}$. *Then for fixed $p_t \in (0, 1)$ and $\tau \to 0$, the log-ratio bounds are*

$$\exp(B_{+,t}^{\mathrm{part}}) = \frac{1}{p_t} - \frac{1 - p_t}{p_t^2}e^{-1/\tau} + O(e^{-2/\tau})$$

$$B_t^{\mathrm{part}} = \frac{1}{\tau} - \log\frac{1}{p_t} + o(1).$$

The proofs follow from Proposition B.1 in Section B.2. By Propositions 4.9 and 4.10, when $\tau$ is small, the factors in PMD-PART are worse than PMD-MEAN, and the gap is significant when $p_t$ is small.

### 4.2.3. TARGET ESTIMATION ERROR

It remains to compare the target estimation error.

**Proposition 4.11** (Target estimation error for binary rewards). *Assume $r(y) \in \{0, 1\}$ and define $p_t := \mathbb{E}_{y \sim \pi_t}[r(y)]$. Let $p_{-i} := \frac{1}{n-1}\sum_{j \neq i} r(y_j)$ and $a := e^{1/\tau} - 1$, then $Z_t = 1 + ap_t$. Fix $\delta \in (0, 1)$ and define*

$$\varepsilon_n(p_t, \delta) := \sqrt{\frac{2p_t(1 - p_t)\log(4n/\delta)}{n - 1}} + \frac{2\log(4n/\delta)}{3(n - 1)}.$$

*Then with probability at least $1 - \delta$, the LOO mean satisfies $\max_i |p_{-i} - p_t| \leq \varepsilon_n(p_t, \delta)$, and consequently:*

*(a) (PMD-MEAN) Assume additionally that $p_t \in (0, 1)$ is fixed and $\tau > 0$ is sufficiently small. For $\widetilde{s}_{-i}^\star(y_i)$ in (17) and $s^\star(y) = \log \frac{\pi_{t+1}^{\mathrm{mean}}(y)}{\pi_t(y)}$,*

$$\overline{\Delta}^2 \lesssim \frac{\varepsilon_n(p_t, \delta)^2}{\tau^2} + \frac{p_t(1 - p_t)^2}{\tau^2}. \tag{25}$$

*(b) (PMD-PART) For $\widetilde{s}_{-i}^\star(y_i)$ in (18) and $s^\star(y) = \log\frac{\pi_{t+1}^{\mathrm{part}}(y)}{\pi_t(y)}$,*

$$\overline{\Delta}^2 \leq \left(\min\left\{\frac{a \cdot \varepsilon_n(p_t, \delta)}{1 + a(p_t - \varepsilon_n(p_t, \delta))_+}, \frac{1}{\tau}\right\}\right)^2. \tag{26}$$

The proof is provided in Section C.5. By Proposition 4.11, the target estimation error of PMD-MEAN consists of two parts. One part scales as $O\left(\frac{p_t(1 - p_t)}{\tau^2 n}\right)$, while the other part $O\left(\frac{p_t(1 - p_t)^2}{\tau^2}\right)$ is irreducible due to the systematic mismatch between the estimated target $\widetilde{s}_{-i}^\star$ the ideal target $s^\star$ that contains the Lambert-$W$ term. In particular, for positive actions, the estimated target is systematically larger than the ideal target, resulting in more aggressive improvement on these actions. Meanwhile, for negative actions, the estimated target is closer to the ideal, thus inherits the more conservative shrinkage at rate $\exp(-p_t/\tau)$.

For PMD-PART, there is a critical regime where $n = \Theta(\frac{1}{p_t})$. When $n$ is smaller than this threshold, the bound almost

scales as $O\big(\min\{\frac{1}{\tau^2}, \frac{e^{2/\tau}p_t}{n}\}\big)$, which is larger than the error of PMD-MEAN. When $n$ exceeds the threshold, the bound scales as $O\big(\frac{1-p_t}{p_t n}\big)$. This suggests that for PMD-PART, the rollout sample size should be large enough, especially at the early phase of training where $p_t$ is small.

Figures 3 and 6 show simulations of target estimation errors, where PMD-PART suffers from larger error when $p_t$ and the rollout sample size $n$ are small. This explains that while PMD-PART has a faster ideal convergence rate (almost in one step for small $\tau$), it can be very unstable in practice when the rollouts are limited. Meanwhile, PMD-MEAN suffers less from target estimation error in this regime.

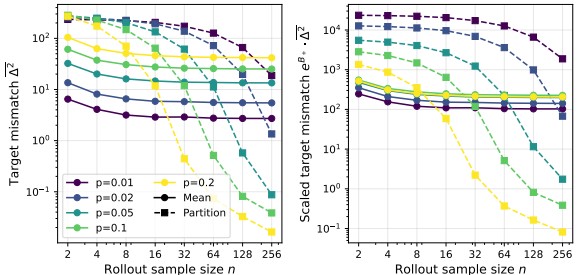

*Figure 3.* Target estimation error of PMD-MEAN and PMD-PART under $\tau = 0.05$ and $p_t$ ranges from 0.01 to 0.2. Left: the target estimation error $\overline{\Delta^2}$. Right: The scaled estimation error with corresponding prefactor $e^{B_+}$ in (22). The plot shows the average from 100 random seeds. When the rollout sample size $n$ is small, the error of PMD-PART is much larger for small $p_t$.

#### 4.2.4. REFINED ANALYSIS FOR PMD-MEAN

The non-vanishing term in Proposition 4.11 is a feature of the pointwise target comparison analysis, not an intrinsic excess risk floor for PMD-MEAN. In fact, the minimizer of the empirical loss (16) recovers the ideal target policy in this limit, as the constraint $\mathbb{E}_{\pi_t}[e^{s_\pi}] = 1$ pulls back the log-ratios from exactly fitting the advantages. We provide a refined analysis that eliminates the error floor.

**Lemma 4.12** (Refined analysis for PMD-MEAN). *Suppose Assumptions 4.1 to 4.4 hold, $r(y) \in \{0,1\}$ and define $p_t := \mathbb{E}_{y \sim \pi_t}[r(y)]$. Assume further that $p_t \in (0,1)$ is fixed and $\tau > 0$ is sufficiently small, as in Proposition 4.9. Let $\varepsilon_n(p_t, \delta)$ be as in Proposition 4.11. Then for any $\delta \in (0,1)$, with probability at least $1 - \delta$,*

$$\mathcal{L}_t(\widehat{\pi}_{t+1}) \lesssim \frac{\log(|\Pi|/\delta)}{\tau^2 n} + \epsilon_{\mathrm{opt}} + \frac{p_t \cdot \varepsilon_n(p_t, \delta) + \varepsilon_n(p_t, \delta)^2}{\tau^2}.$$

The complete proof is provided in Appendix D. Lemma 4.12 shows that PMD-MEAN now shares the same $O\big(\frac{\log(|\Pi|/\delta)}{\tau^2 n}\big)$ term as PMD-PART in (20) (as $B_t^{\mathrm{part}} = O\big(\frac{1}{\tau}\big)$), while the $\overline{\Delta^2}$ term now becomes $O\big(\frac{p_t}{\tau^2}\sqrt{\frac{p_t}{n}} + \frac{1}{\tau^2 n^2}\big)$, which vanishes as $n \to \infty$. This is better than the $O\big(\frac{1}{\tau^2}\big)$ error of PMD-PART in the small $n$ and $p_t$ regime. On the other hand, when

$n = \Omega\big(\frac{1}{p_t}\big)$, the error of PMD-PART is $O(1)$, while the error in PMD-MEAN is $O\big(\frac{p_t^2}{\tau^2}\big)$. This suggests an adaptive regularization scheme of PMD-MEAN that scales $\tau$ with per-prompt pass rate $p_t$, which we leave for future investigation.

## 5. Experiments

We conduct experiments on math reasoning RL to validate the practical performance of PMD. Our implementation is based on verl (Sheng et al., 2025).

We train on the DAPO-Math-17k dataset (Yu et al., 2025) with base models Qwen2.5-7B (Yang et al., 2024) and Qwen3-30B-A3B-Base (Yang et al., 2025). The 7B models are trained for 495 global steps (15 epochs), while the 30B models are trained for 300 global steps. We apply the same prompt formatting as in Yu et al. (2025). The reward is binary $\pm 1$ based on the answer correctness only. This is an affine rescaling of the binary reward in theory, and the additive shift cancels in both PMD-MEAN and PMD-PART. Thus the experimental losses with temperature $\tau$ correspond to the theory with effective temperature $\tau/2$.

We set the global batch size as 512 prompts with group size 16 and sampling temperature 1 for rollout. The maximum response length is 8192 for Qwen2.5-7B and 20480 for Qwen3-30B-A3B-Base. We train with a mini-batch size of 32 prompts (512 sequences) and learning rate $1 \times 10^{-6}$.

We evaluate on AIME 2024 and AIME 2025. For each problem, we sample 32 solutions and report the average score. We mainly use GRPO (Shao et al., 2024) as the baseline. For efficiency comparison, we also include the on-policy gradient by setting the global batch size (rollout prompts) as 32 so that it equals the mini-batch size. More implementation details and additional experiments with more evaluations are provided in Section A.

### 5.1. Main Results

The main results are shown in Table 1. In our staleness-16 setting, PMD-MEAN obtains higher Avg@32 than the GRPO baseline on both evaluated model families: $\tau = 0.005$ gives +2.6% AIME24 and +9.0% AIME25 absolute gains on the 7B model, and $\tau = 0.1$ gives +14.3% AIME24 and +9.3% AIME25 absolute gains on the 30B MoE model.

**Efficiency from larger global batch size.** As shown in Tables 1 and 2, compared to the on-policy gradient with staleness 1, the off-policy PMD-MEAN achieves comparable performance with $4.6\times$ speedup by leveraging a larger global rollout batch size that amortizes the inference cost.

**Stability.** As shown in Figure 4, PMD-MEAN remains stable during training, while PMD-PART is highly unstable and could collapse even with a much larger $\tau$.

*Table 1.* Overall evaluation scores (Avg@32). Staleness indicates the number of ministeps using rollouts from the same old policy.

| METHOD ($\tau$) | STALENESS | AIME 24 | AIME 25 | AVERAGE |
|---|---|---|---|---|
| **QWEN2.5-7B** | | | | |
| GRPO (-) | 16 | 17.08 | 10.52 | 13.80 |
| ON-POLICY (-) | 1 | 18.65 | 18.33 | 18.49 |
| PMD-MEAN (0.005) | 16 | 19.69 | **19.48** | **19.58** |
| PMD-MEAN (0.01) | 16 | 17.60 | 17.50 | 17.55 |
| PMD-MEAN (0.02) | 16 | **22.50** | 16.67 | **19.58** |
| **QWEN3-30B-A3B-BASE** | | | | |
| GRPO (-) | 16 | 36.56 | 27.92 | 32.24 |
| PMD-MEAN (0.01) | 16 | 50.00 | 35.10 | 42.55 |
| PMD-MEAN (0.1) | 16 | **50.83** | **37.19** | **44.01** |

*Table 2.* Comparing efficiency of on-policy gradient and PMD-MEAN. Timing is in milliseconds per token. While the actor update cost is comparable, a larger global batch size (high staleness) amortizes the inference cost and reduces overall training time.

| METHOD | OVERALL (MS/TOKEN) | GENERATION | ACTOR UPDATE |
|---|---|---|---|
| ON-POLICY | 0.0569 | 0.0512 | **0.0057** |
| PMD-MEAN | **0.0126** | **0.0062** | 0.0064 |

**Policy ratios.** We record the log policy ratios between the actor policy $\pi_\theta$ in the last mini-step and the old rollout policy $\pi_t$ of that global step, using it as an approximation of $\log \frac{\pi_{t+1}}{\pi_t}$. As shown in Figure 5, the trend validates our theory in Section 4.2.2 that the policy decrease in PMD-MEAN is weaker than PMD-PART, and becomes stronger as training proceeds and accuracy improves.

**Beyond standard GRPO.** Standard GRPO faces stability issues in training large MoE models. We further compare PMD-MEAN with GSPO (Zheng et al., 2025), which is an advanced variant of GRPO that incorporates sequence-level importance sampling (IS) with clipping and geometric mean normalization to resolve this MoE stability issue and achieves superior performance to GRPO. As shown in Table 3, PMD-MEAN is stronger than GSPO on Qwen2.5-7B and matches GSPO on average for Qwen3-30B-A3B-Base, with different datasets favoring different methods.

## 6. Related Work

**RL for Post-Training of LLMs.** Contemporary LLM post-training and alignment frameworks predominantly utilize reinforcement learning with human or AI feedback (RLHF/RLAIF, Ziegler et al. 2019; Ouyang et al. 2022; Bai et al. 2022) or verifiable rewards (RLVR, Lambert et al. 2024). This paradigm has demonstrated particular efficacy for mathematical reasoning, coding, and logical tasks, subsequently inspiring large-scale RL methodologies and architectural designs for increasingly complex agentic capabilities (OpenAI, 2024; Guo et al., 2025; Google, 2025; Team et al., 2025a).

Policy gradient methods (Williams, 1992; Sutton et al., 1999), especially TRPO and PPO (Schulman et al., 2015; 2017), have established themselves as foundational ap-

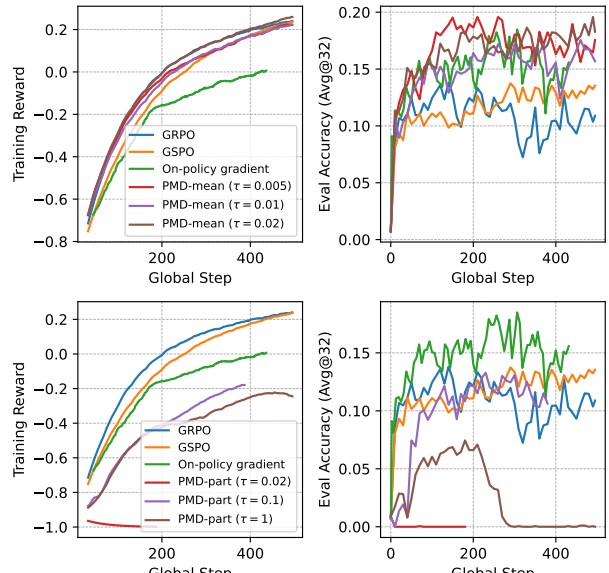

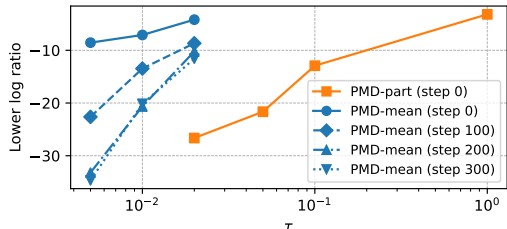

*Figure 4.* Training curves (smoothed) of PMD-MEAN (upper) and PMD-PART (lower) with baselines for Qwen2.5-7B on DAPO-Math-17k (left) and the averaged evaluation accuracy on AIME 2024 and AIME 2025 (right). The global step of on-policy gradient is divided by 16 to match other algorithms.

*Figure 5.* The minimum of log-ratios $\log \frac{\pi_{t+1}}{\pi_t}$ in PMD-MEAN and PMD-PART, estimated from the last update mini-batch.

proaches in reinforcement learning. However, within LLM post-training contexts, maintaining parameterized value networks (critic models) introduces substantial estimation biases and computational overhead. To mitigate these limitations, recent methods such as GRPO (Shao et al., 2024) and RLOO (Ahmadian et al., 2024) eliminate the critic component and instead leverage group-based relative baselines estimated from multiple Monte Carlo samples per prompt. More sophisticated approaches, such as DAPO (Yu et al., 2025), further refine performance through advanced optimization techniques.

These methods fundamentally depend on sampling distributions that closely match the current policy distribution, necessitating off-policy correction mechanisms to maintain training stability. While PPO/GRPO implement token-level importance sampling (IS) with clipping, more recent algorithms such as GSPO (Zheng et al., 2025) and CISPO (Chen et al., 2025) employ sequence-level IS or detached clipping IS to enhance stability when training mixture-of-

*Table 3.* Comparison of PMD-MEAN and GSPO (Avg@32).

| METHOD | MODEL | AIME 24 | AIME 25 | AVERAGE |
|---|---|---|---|---|
| GSPO | 7B | 15.52 | 11.98 | 13.75 |
| PMD-MEAN | 7B | **19.69** | **19.48** | **19.58** |
| GSPO | 30B | **53.33** | 34.58 | 43.96 |
| PMD-MEAN | 30B | 50.83 | **37.19** | **44.01** |

expert (MoE) models. Additional research addresses the training-inference distribution mismatch at the infrastructure level, proposing methodological refinements including truncated IS (Yao et al., 2025) and masked IS (Liu et al., 2025). Although partially effective, these techniques substantially increase algorithmic complexity and incorporate numerous empirical adjustments that resist theoretical analysis. In contrast, our investigation focuses on PMD-MEAN, a minimalist PMD-style algorithm that offers greater analytical transparency while delivering competitive empirical performance.

**Policy Mirror Descent.** The mirror descent framework (Nemirovsky & Yudin, 1983) provides a classical formulation for policy optimization in reinforcement learning (Geist et al., 2019; Tomar et al., 2022), with extensive literature analyzing its theoretical iteration and sample complexities (Xiao, 2022; Zhan et al., 2023; Lan, 2023; Yuan et al., 2023; Alfano et al., 2023; Xu et al., 2024). However, these analyses predominantly address tabular settings or function approximation scenarios with abundant samples, rather than the practical constraints of LLM post-training where rollouts are necessarily limited.

Our work establishes a connection between practical LLM post-training methodologies and the choice of Bregman divergence, demonstrating that PMD-MEAN implicitly optimizes an adaptive mixed KL–$\chi^2$ regularizer. This mixed regularization approach shares conceptual similarities with $\chi^2$PO (Huang et al., 2024), which employs mixed KL–$\chi^2$ divergence to mitigate overfitting in KL-regularized DPO (Rafailov et al., 2023) under distribution shift conditions.

This mechanism is distinct from other recent implicit regularization phenomena in RL. Liu et al. (2024) show that coupling preference optimization with an SFT loss yields an implicit adversarial regularizer for RLHF; Qiao et al. (2026) show that squared TD objectives can induce harmful cross-covariance in offline RL and propose clustered cross-covariance control; and Xu et al. (2025) use implicit value regularization around a reference policy to unify online, offline, and offline-to-online RL. In contrast, PMD-MEAN's regularizer appears directly in the online mirror-descent subproblem because the log-partition term is replaced by a rollout mean under the policy-normalization constraint.

Our empirical study focuses on critic-free RLVR methods in the same stale-rollout training regime. We do not empirically sweep alternative $f$-divergence regularizers or regression-based PMD variants. Exploring adaptive $\tau$ sched-

ules, pass-rate-dependent regularization, and other divergence choices is a natural next step.

The literature offers various regression-based approaches to approximate the ideal KL solution in PMD-PART. Richemond et al. (2024) propose incorporating a value network to estimate the log-partition function, which dates back to Nachum et al. (2017). Gao et al. (2024) develop a technique for fitting pairwise relative rewards that eliminates the partition term entirely. Bartoldson et al. (2025) approximate the log-partition term in the loss using the group average of all other terms (with stop-grad). In contrast, PMD-MEAN (Team et al., 2025b) implements a simpler strategy by directly approximating the log-partition function with the mean reward. Our analysis focuses on characterizing the theoretical properties of this practical approximation and establishing its mathematical foundations.

## 7. Conclusion

This paper presents a comprehensive analysis of PMD-MEAN, a practical algorithm for large-scale LLM post-training that has been deployed in leading language models. The analysis provides an exact characterization of the algorithm's population update through the Lambert-$W$ function, establishing its mathematical equivalence to solving mirror descent subproblems with an adaptive mixed KL–$\chi^2$ regularizer. This theoretical framework reveals a concrete mechanism underlying the algorithm's stability: the induced $\chi^2$ term systematically constrains large probability ratio changes, effectively preventing overly aggressive policy updates that often lead to training instability.

The investigation deliberately focuses on the principled form of PMD-MEAN to enable clearer theoretical analysis and understanding. Advanced techniques such as oversampling strategies and importance sampling corrections for addressing training/inference engine mismatches could potentially enhance PMD-MEAN's performance further, and these directions are reserved for future research. By elucidating the fundamental mathematical properties of PMD-MEAN, this work contributes to the development of theoretically grounded yet practically effective RL algorithms for LLM post-training, potentially enabling simpler, more robust, and more scalable approaches to this increasingly critical task.

## Impact Statement

This paper presents work whose goal is to advance the field of Machine Learning. There are many potential societal consequences of our work, none of which we feel must be specifically highlighted here.

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

# A. Experimental Details

We train on the DAPO-Math-17k[1] dataset (deduplicated). The base models include Qwen2.5-7B[2] and Qwen3-30B-A3B-Base[3].

## A.1. Prompt Template

Our prompt template follows Yu et al. (2025), with all questions `problem_statement` processed in the following form.

```
Chain-of-Thought (CoT) Prompt Template

Solve the following math problem step by step. The last line of your response
    should be of the form Answer: $Answer (without quotes) where $Answer is the
    answer to the problem.

{problem_statement}

Remember to put your answer on its own line after "Answer:".
```

## A.2. Hyperparameters

We summarize key hyperparameters in Table 4.

Table 4. Key hyperparameters for 7B dense model and 30B MoE model experiments.

| Parameter | 7B Dense | 30B MoE |
|---|---|---|
| trainer.nnodes | 4 | 8 |
| trainer.n_gpus_per_node | 8 | 8 |
| distributed strategy | FSDP | Megatron |
| model.path | Qwen/Qwen2.5-7B | Qwen/Qwen3-30B-A3B-Base |
| data.train_batch_size (prompts) | 512 | 512 |
| data.gen_batch_size (prompts) | 512 | 512 |
| data.max_prompt_length | 2048 | 2048 |
| data.max_response_length | 8192 | 20480 |
| rollout.name | vLLM | vLLM |
| rollout.n (group size) | 16 | 16 |
| rollout.temperature | 1.0 | 1.0 |
| rollout.top_p | 1.0 | 1.0 |
| rollout.max_model_len | 10240 | 22528 |
| val_kwargs.n (avg@k) | 32 | 32 |
| val_kwargs.temperature | 1.0 | 1.0 |
| val_kwargs.top_p | 0.7 | 0.7 |
| actor.ppo_epochs | 1 | 1 |
| actor.ppo_mini_batch_size (prompts) | 32 | 32 |
| actor.clip_ratio_low / high (GRPO) | 0.2 / 0.2 | 0.2 / 0.2 |
| actor.clip_ratio_low / high (GSPO) | 3e-4 / 4e-4 | 3e-4 / 4e-4 |
| optim.lr | 1e-6 | 1e-6 |
| optim.betas | [0.9, 0.999] | [0.9, 0.999] |
| optim.weight_decay | 0.01 | 0.01 |
| grad clip | 1.0 | 1.0 |
| tensor_model_parallel_size | 1 | 2 |
| pipeline_model_parallel_size | 1 | 2 |
| expert_model_parallel_size | N/A | 8 |

---

[1] https://huggingface.co/datasets/BytedTsinghua-SIA/DAPO-Math-17k
[2] https://huggingface.co/Qwen/Qwen2.5-7B
[3] https://huggingface.co/Qwen/Qwen3-30B-A3B-Base

## A.3. Implementation Details

Our implementation is based on verl[4] (Sheng et al., 2025). For GRPO and GSPO, we disable explicit KL penalty with respect to the base reference model, following the DAPO recipe (Yu et al., 2025). For each prompt $x$, we sample $K$ responses $y_1, \ldots, y_K \sim \pi_t(\cdot \mid x)$ from the old policy $\pi_t(\cdot \mid x) := \pi_{\theta_t}(\cdot \mid x)$, and get rewards $r_i := r(x, y_i) \in \{-1, +1\}$ based on the correctness of answers. Let $\pi_\theta$ be the trainable policy. We have $\log \pi_\theta(y \mid x) = \sum_{j=1}^{|y|} \log \pi_\theta(y_j \mid x, y_{<j})$.

Define the token probability ratio and geometric normalized sequence probability ratio as follows:

$$\rho_{i,j}(\theta) := \frac{\pi_\theta(y_{i,j} \mid x, y_{i,<j})}{\pi_t(y_{i,j} \mid x, y_{i,<j})}, \quad s_i(\theta) := \left( \frac{\pi_\theta(y_i \mid x)}{\pi_t(y_i \mid x)} \right)^{\frac{1}{|y_i|}}.$$

Moreover, define the following advantages:

$$\widehat{A}_i^{\mathrm{grpo}} := \frac{r_i - \mathrm{mean}(r_1, \ldots, r_K)}{\mathrm{std}(r_1, \ldots, r_K)}, \quad \widehat{A}_i^{\mathrm{loo}} := r_i - \frac{1}{K-1} \sum_{j \neq i} r_j, \quad \widehat{A}_i^{\mathrm{part}} := r_i - \tau \log \left( \frac{1}{K-1} \sum_{j \neq i} e^{r_j/\tau} \right).$$

The GRPO (Shao et al., 2024) loss is defined as

$$\mathcal{L}_{\mathrm{GRPO}}(\theta) = -\mathbb{E}_{x \sim \mathcal{D}} \mathbb{E}_{y_1, \ldots, y_K \sim \pi_t(\cdot \mid x)} \left[ \frac{1}{K} \sum_{i=1}^{K} \frac{1}{|y_i|} \sum_{j=1}^{|y_i|} \min \left( \rho_{i,j}(\theta) \widehat{A}_i^{\mathrm{grpo}}, \mathrm{clip}(\rho_{i,j}(\theta), 1 - \epsilon, 1 + \epsilon) \widehat{A}_i^{\mathrm{grpo}} \right) \right],$$

where $\epsilon = 0.2$ and we discard the KL penalty term.

When the global batch size is set to be the same as the mini-batch size, GRPO reduces to the on-policy gradient with advantage estimator $\widehat{A}_i^{\mathrm{grpo}}$. Empirically, we find that using $\widehat{A}_i^{\mathrm{loo}}$ yields similar performance, and thus we use the (length-normalized) RLOO (Ahmadian et al., 2024) loss for on-policy gradient experiments.

$$\mathcal{L}_{\mathrm{RLOO}}(\theta) = -\mathbb{E}_{x \sim \mathcal{D}} \mathbb{E}_{y_1, \ldots, y_K \sim \pi_t(\cdot \mid x)} \left[ \frac{1}{K} \sum_{i=1}^{K} \frac{1}{|y_i|} \widehat{A}_i^{\mathrm{loo}} \log \pi_\theta(y_i \mid x) \right].$$

The GSPO (Zheng et al., 2025) loss is defined as follows:

$$\mathcal{L}_{\mathrm{GSPO}}(\theta) = -\mathbb{E}_{x \sim \mathcal{D}} \mathbb{E}_{y_1, \ldots, y_K \sim \pi_t(\cdot \mid x)} \left[ \frac{1}{K} \sum_{i=1}^{K} \min \left( s_i(\theta) \widehat{A}_i^{\mathrm{grpo}}, \mathrm{clip}(s_i(\theta), 1 - \epsilon_{\mathrm{low}}, 1 + \epsilon_{\mathrm{high}}) \widehat{A}_i^{\mathrm{grpo}} \right) \right],$$

where $\epsilon_{\mathrm{low}} = 3 \times 10^{-4}$ and $\epsilon_{\mathrm{high}} = 4 \times 10^{-4}$ as suggested.

We implement PMD-MEAN (Team et al., 2025b) and PMD-PART using the following losses.

$$\mathcal{L}_{\mathrm{mean}}(\theta) = \mathbb{E}_{x \sim \mathcal{D}} \mathbb{E}_{y_1, \ldots, y_K \sim \pi_t(\cdot \mid x)} \left[ \frac{1}{K} \sum_{i=1}^{K} \frac{\tau}{|y_i|} \left( \log \frac{\pi_\theta(y_i \mid x)}{\pi_t(y_i \mid x)} - \frac{\widehat{A}_i^{\mathrm{loo}}}{\tau} \right)^2 \right],$$

$$\mathcal{L}_{\mathrm{part}}(\theta) = \mathbb{E}_{x \sim \mathcal{D}} \mathbb{E}_{y_1, \ldots, y_K \sim \pi_t(\cdot \mid x)} \left[ \frac{1}{K} \sum_{i=1}^{K} \frac{\tau}{|y_i|} \left( \log \frac{\pi_\theta(y_i \mid x)}{\pi_t(y_i \mid x)} - \frac{\widehat{A}_i^{\mathrm{part}}}{\tau} \right)^2 \right].$$

The factor $\tau$ in the loss ensures the gradient norm does not differ too much when tuning $\tau$, and length normalization is consistent with the loss aggregation mode in other methods.

## A.4. Additional Experiments

To further test robustness under broader evaluation coverage and stronger off-policyness, we run additional experiments with Qwen3-8B-Base[5]. These runs are based on slime[6] instead of verl used in the main experiments. The training data and reward remains unchanged.

---

[4]https://github.com/verl-project/verl
[5]https://huggingface.co/Qwen/Qwen3-8B-Base
[6]https://github.com/THUDM/slime

In all additional experiments, each training prompt is paired with 32 sampled responses, and the actor is updated with mini-batches of 128 sampled trajectories. In the staleness-16 setting, each rollout batch contains 64 prompts, producing 2048 sampled trajectories. These trajectories are then reused for 16 consecutive actor-update steps before collecting new rollouts. In the staleness-64 setting, each rollout batch contains 256 prompts, producing 8192 sampled trajectories, which are reused for 64 actor-update steps. The staleness-16 runs use 250 rollout batches, while the staleness-64 runs use 64 rollout batches, so both settings perform roughly 4K actor-update steps, or equivalently, one epoch of DAPO-Math-17k dataset in total. We set $\tau = 0.01$ for PMD-MEAN. Other hyperparameters such as the learning rate follow the main experiment.

We evaluate on AIME24[7], AIME25[8], AIME26[9], and AMC23[10]. For each problem, we sample 32 solutions and report Avg@32, with maximum response length 16384.

*Table 5.* Additional best-checkpoint evaluation scores (Avg@32) for Qwen3-8B-Base. Staleness indicates the number of ministeps using rollouts from the same old policy. Average is computed over AIME24, AIME25, AIME26, and AMC23.

| METHOD ($\tau$) | STALENESS | AIME 24 | AIME 25 | AIME 26 | AMC 23 | AVERAGE |
|---|---|---|---|---|---|---|
| PMD-MEAN (0.01) | 16 | **34.9** | 29.2 | 26.9 | **81.0** | **43.0** |
| GRPO (-) | 16 | 26.8 | 22.2 | 23.2 | 70.9 | 35.8 |
| GSPO (-) | 16 | 32.7 | **29.4** | **28.3** | 77.9 | 42.1 |
| PMD-MEAN (0.01) | 64 | **34.5** | **28.3** | **26.0** | **79.2** | **42.0** |
| GRPO (-) | 64 | 27.1 | 24.2 | 24.3 | 72.1 | 36.9 |
| GSPO (-) | 64 | 27.7 | 23.3 | 23.3 | 74.1 | 37.1 |

As shown in Table 5, PMD-MEAN remains competitive under this separate Slime-based setup. At staleness 16, PMD-MEAN achieves the best average score, slightly outperforming GSPO and improving over GRPO by 7.2 points on average. At staleness 64, PMD-MEAN maintains a similar average score and outperforms both GRPO and GSPO by about 5 points. These results are consistent with the main experiments: the PMD-MEAN update remains robust when the rollout policy is stale, while avoiding the severe degradation observed for GRPO in the high-staleness setting.

### A.5. Supplementary Results

We provide supplementary experimental results in Figures 6 to 8.

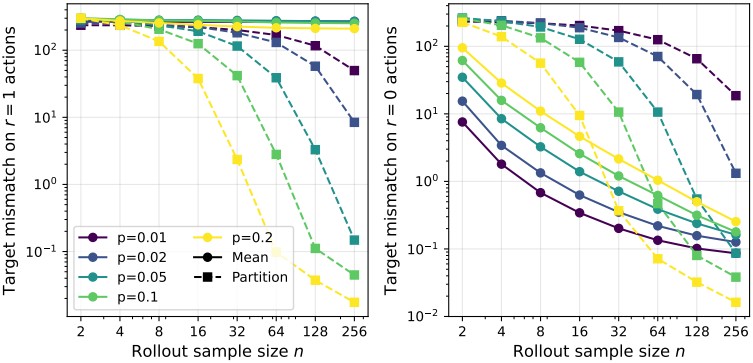

*Figure 6.* Target estimation error for positive and negative actions in PMD-MEAN and PMD-PART under $\tau = 0.05$ and $p_t$ ranges from 0.01 to 0.2. Left: positive actions. Right: negative actions. The plot shows the average from 100 random seeds. The error in PMD-MEAN mainly comes from a systematic mismatch between positive targets.

---

[7] https://huggingface.co/datasets/MathArena/aime_2024_I and https://huggingface.co/datasets/MathArena/aime_2024_II
[8] https://huggingface.co/datasets/MathArena/aime_2025
[9] https://huggingface.co/datasets/MathArena/aime_2026
[10] https://huggingface.co/datasets/weqweasdas/amc23

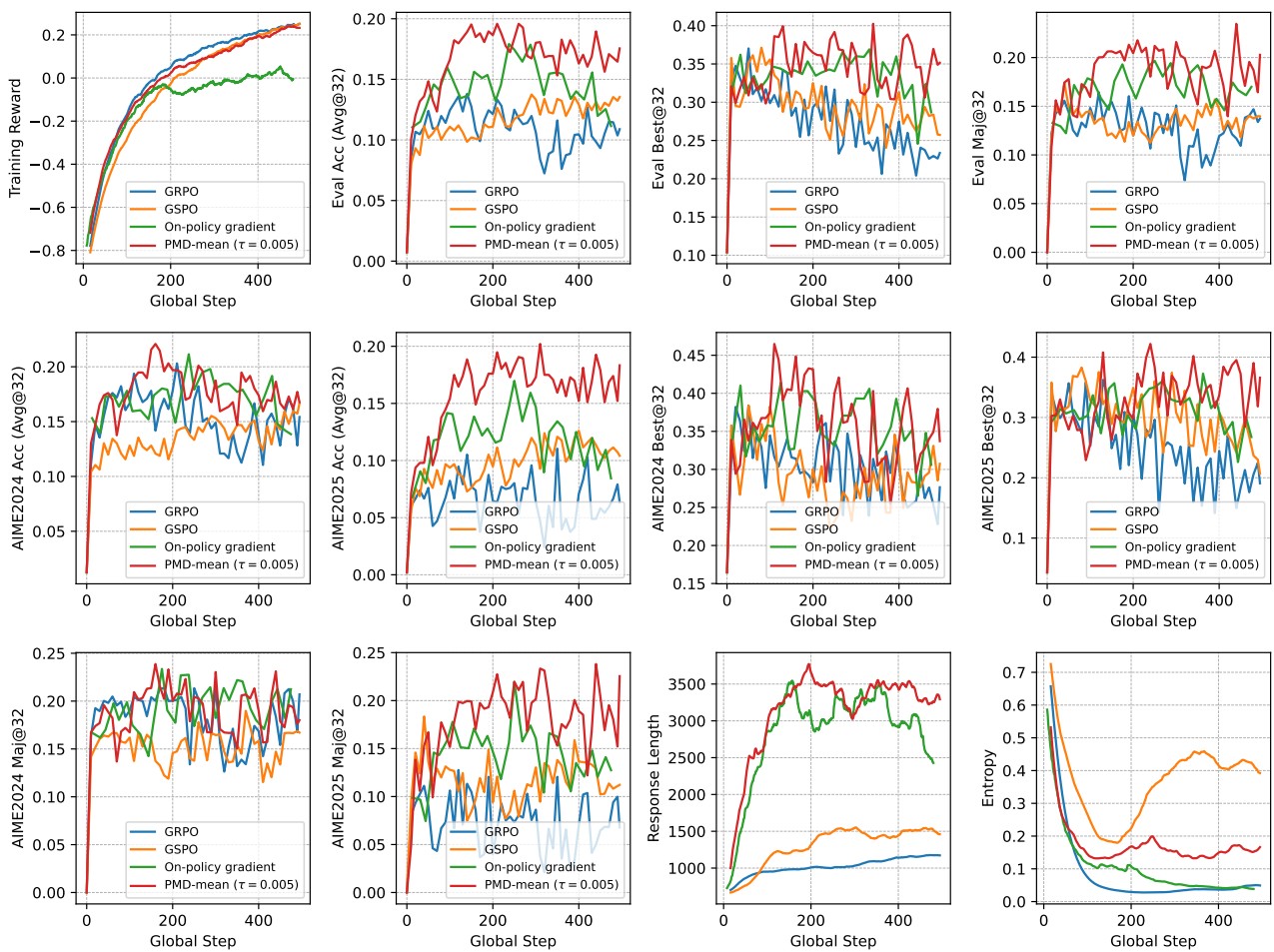

*Figure 7.* Qwen2.5-7B training results for 15 epochs (495 global steps). Training reward, response length, and entropy are smoothed by EMA with an effective window size of 50. PMD-MEAN achieves stronger or competitive scores in these runs, not only in Pass@1 (measured in Avg@32) but also Pass@32 and Maj@32 (accuracy of majority voting answer).

## B. Missing Proofs in Section 3

### B.1. Proof of Theorem 3.1

*Proof of Theorem 3.1.* Define $u(y) = \log \frac{\pi(y)}{\pi_t(y)}$. We multiply $\mathcal{L}_{\text{mean}}$ by the positive constant $\tau^2$, which does not change the minimizer, and write the Lagrangian in the policy space as

$$L(u, \lambda) = \frac{1}{2} \sum_{y \in \mathcal{Y}} \pi_t(y)(\Delta_y - \tau u(y))^2 + \lambda \cdot \left( \sum_{y \in \mathcal{Y}} \pi_t(y) e^{u(y)} - 1 \right),$$

The KKT conditions yield

$$\begin{cases} -\tau \pi_t(y)(\Delta_y - \tau u(y)) + \lambda \pi_t(y) e^{u(y)} = 0, \ \forall y \in \mathcal{Y}, \\ \sum_{y \in \mathcal{Y}} \pi_t(y) e^{u(y)} = 1. \end{cases}$$

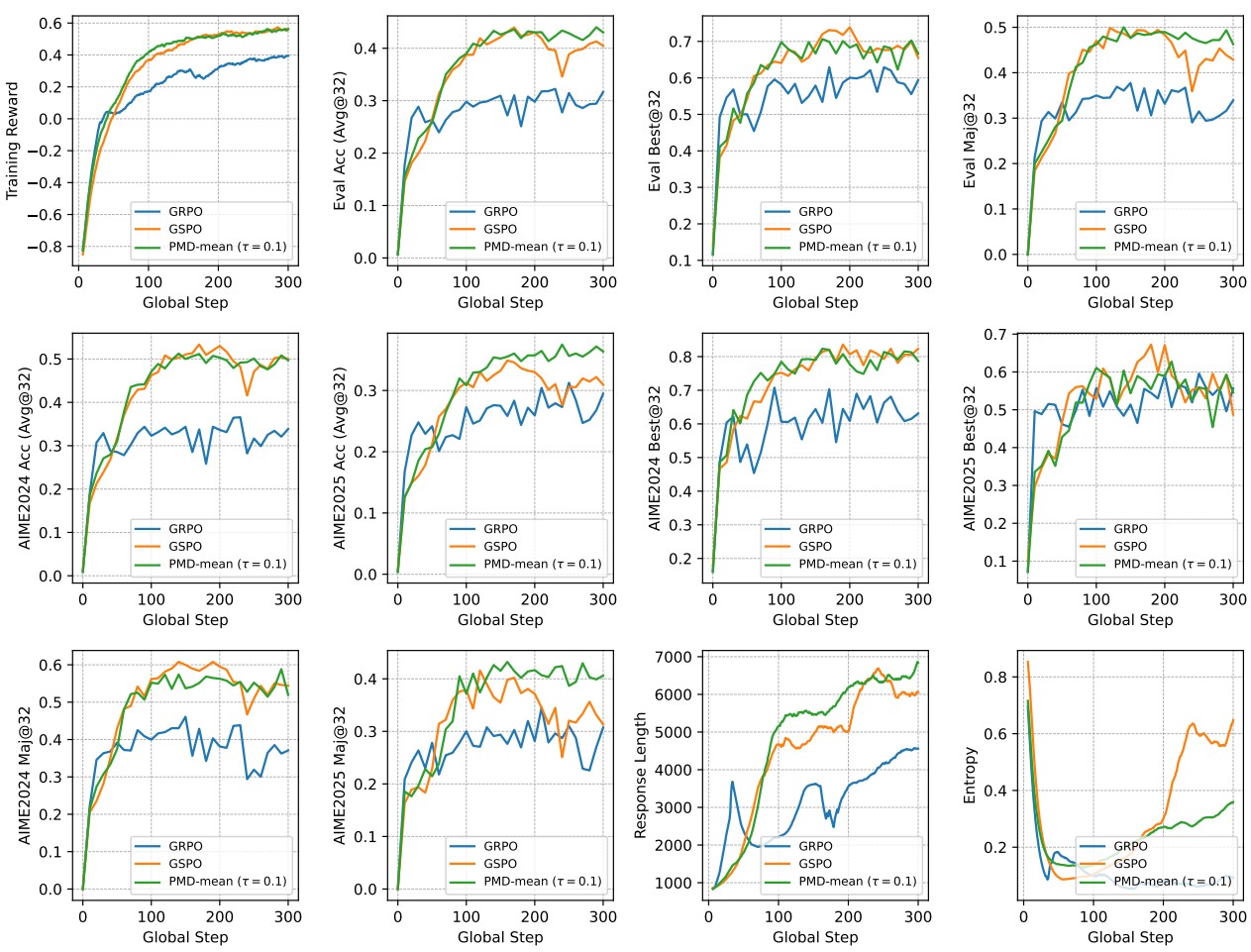

*Figure 8.* Qwen3-30B-A3B-Base training results for 300 global steps. Training reward, response length, and entropy are smoothed by EMA with an effective window size of 20.

Assume $\pi_t(y) > 0$ for all $y \in \mathcal{Y}$ (which is true for LLMs without top-p/top-k constraints), then any stationary point of $\mathcal{L}_{\text{mean}}$ should satisfy

$$
\begin{aligned}
-\tau(\Delta_y - \tau u(y)) + \lambda e^{u(y)} = 0 &\iff \tau(\Delta_y - \tau u(y))e^{-u(y)} = \lambda \\
&\iff \left(\frac{\Delta_y}{\tau} - u(y)\right) e^{\frac{\Delta_y}{\tau} - u(y)} = \frac{\lambda}{\tau^2} e^{\frac{\Delta_y}{\tau}} \\
&\iff \frac{\Delta_y}{\tau} - u(y) = W\left(\frac{\lambda}{\tau^2} e^{\frac{\Delta_y}{\tau}}\right) \\
&\iff u(y) = \frac{\Delta_y}{\tau} - W\left(\frac{\lambda}{\tau^2} e^{\frac{\Delta_y}{\tau}}\right).
\end{aligned}
\tag{27}
$$

For each $\lambda \geq 0$, define

$$
u_\lambda(y) := \frac{\Delta_y}{\tau} - W\left(\frac{\lambda}{\tau^2} e^{\frac{\Delta_y}{\tau}}\right), \quad F(\lambda) = \mathbb{E}_{\pi_t}\left[e^{u_\lambda(y)}\right].
$$

Since $W$ is continuous and strictly increasing on $[0, \infty)$, $F$ is continuous and strictly decreasing in $\lambda$. Moreover,

$$
F(0) = \mathbb{E}_{\pi_t}\left[e^{\Delta_y/\tau}\right] \geq \exp\left(\mathbb{E}_{\pi_t}[\Delta_y]/\tau\right) = 1
$$

by Jensen's inequality, and $F(\lambda) \to 0$ as $\lambda \to \infty$ because $W(z) \to \infty$ as $z \to \infty$. Hence, there is a unique $\lambda^\star \geq 0$ such that $F(\lambda^\star) = 1$. If $F(0) = 1$, then $\Delta_y \equiv 0$ and $\lambda^\star = 0$. Otherwise, we have $\lambda^\star > 0$. This proves the existence and uniqueness of the normalization constant.

For this $\lambda^\star$, $u_{\lambda^\star}$ satisfies the KKT conditions. Moreover, since $\lambda^\star \geq 0$, the Hessian of the Lagrangian with respect to $u$ is diagonal with entries $\pi_t(y)(\tau^2 + \lambda^\star e^{u_{\lambda^\star}(y)}) > 0$. Thus the Lagrangian is strictly convex in $u$. Since the constraint term vanishes on the feasible set, the KKT point is the unique global minimizer of $\mathcal{L}_{\text{mean}}$ over the simplex.

If $\Delta_y \equiv 0$, then $\lambda = 0$ and all bounds are trivial. Hence, in the remainder assume the non-degenerate case so that $\lambda > 0$. To characterize the normalization constant $\lambda$, we invoke several properties of $W$ for all $z \geq 0$: (1) $W(z)$ is concave and monotonically increasing; (2) $W(z) \geq \frac{z}{1+z}$; (3) $e^{W(z)} = \frac{z}{W(z)}$. Moreover, by the feasibility constraint, we have

$$
\begin{aligned}
1 &= \sum_{y \in \mathcal{Y}} \pi_t(y) \frac{\exp\left(\frac{\Delta_y}{\tau}\right)}{\exp\left(W\left(\frac{\lambda}{\tau^2}\exp\left(\frac{\Delta_y}{\tau}\right)\right)\right)} \\
&= \sum_{y \in \mathcal{Y}} \pi_t(y) \frac{\exp\left(\frac{\Delta_y}{\tau}\right) W\left(\frac{\lambda}{\tau^2}\exp\left(\frac{\Delta_y}{\tau}\right)\right)}{\frac{\lambda}{\tau^2}\exp\left(\frac{\Delta_y}{\tau}\right)} \\
&= \frac{\tau^2}{\lambda} \mathbb{E}_{y \sim \pi_t}\left[W\left(\frac{\lambda}{\tau^2}\exp\left(\frac{\Delta_y}{\tau}\right)\right)\right].
\end{aligned}
$$

For convenience, let $x := \lambda/\tau^2 \geq 0$, $A := \mathbb{E}_{\pi_t}[e^{\Delta_y/\tau}]$, $B := \mathbb{E}_{\pi_t}[e^{2\Delta_y/\tau}]$. Then the target is to show

$$
\frac{A(A-1)}{B} \leq x \leq \log A.
$$

For the upper bound, by concavity and Jensen's inequality,

$$
x = \mathbb{E}_{\pi_t}[W(xe^{\Delta_y/\tau})] \leq W(\mathbb{E}[xe^{\Delta_y/\tau}]) = W(xA).
$$

Since $W$ is increasing, above inequality implies

$$
xe^x \leq W(xA)e^{W(xA)} = xA \implies e^x \leq A \implies x \leq \log A,
$$

thus proving the upper bound.

On the other hand, by lower bound on $W$,

$$
\begin{aligned}
x &= \mathbb{E}_{\pi_t}[W(xe^{\Delta_y/\tau})] \\
&\geq \mathbb{E}_{\pi_t}\left[\frac{xe^{\Delta_y/\tau}}{1 + xe^{\Delta_y/\tau}}\right] \\
&\geq \frac{\left(\mathbb{E}_{\pi_t}[xe^{\Delta_y/\tau}]\right)^2}{\mathbb{E}_{\pi_t}[xe^{\Delta_y/\tau}(1 + xe^{\Delta_y/\tau})]} \\
&= \frac{(xA)^2}{xA + x^2 B},
\end{aligned}
$$

where the second inequality is from Cauchy-Schwarz. Solving the inequality yields the lower bound.

For binary rewards $r \in \{0, 1\}$ with $p = \mathbb{E}_{\pi_t}[r(y)]$, we have $\mathbb{E}_{\pi_t}[\Delta_y^2] = \text{Var}(r) = p(1-p)$ and

$$
x = pW(xe^{(1-p)/\tau}) + (1-p)W(xe^{-p/\tau}). \tag{28}
$$

For large $\tau \to \infty$, we have

$$
A = \mathbb{E}_{\pi_t}\left[1 + \frac{\Delta_y}{\tau} + \frac{\Delta_y^2}{2\tau^2} + O(\tau^{-3})\right] = 1 + \frac{p(1-p)}{2\tau^2} + O(\tau^{-3}),
$$

$$
B = \mathbb{E}_{\pi_t}\left[1 + \frac{2\Delta_y}{\tau} + \frac{2\Delta_y^2}{\tau^2} + O(\tau^{-3})\right] = 1 + \frac{2p(1-p)}{\tau^2} + O(\tau^{-3}).
$$

Thus, the lower bounds on $x = \lambda/\tau^2$ yield

$$A(A-1) \leq xB \implies \left(\frac{p(1-p)}{2\tau^2}\right)(1 + O(\tau^{-2})) \leq (1 + O(\tau^{-2}))\frac{\lambda}{\tau^2}$$

$$\implies \frac{p(1-p)}{2} - O(\tau^{-1}) \leq \lambda.$$

On the other hand, the upper bound yields

$$\lambda \leq \tau^2 \log A = \tau^2 \log\left(1 + \frac{p(1-p)}{2\tau^2} + O(\tau^{-3})\right) \leq \tau^2\left(\frac{p(1-p)}{2\tau^2} + O(\tau^{-3})\right) = \frac{p(1-p)}{2} + O(\tau^{-1}).$$

Combine the two bounds, we have $\lambda = \frac{1}{2}p(1-p) + O(\tau^{-1})$ as $\tau \to \infty$.

For small $\tau \to 0$, the bounds (6) are too loose. We define $v = \frac{\lambda}{\tau} = \tau x$ and show that $v \to p(1-p)$. Firstly, the Lambert-$W$ function satisfies

$$\log z - \log\log z \leq W(z) \leq \log z$$

for $z > e$. Moreover, for $z \geq 0$, $W(z) = \frac{z}{e^{W(z)}} \leq z$. Since $e^{-p/\tau}$ decays faster than any polynomial in $\tau^{-1}$, we have

$$0 \leq (1-p)\tau W(xe^{-p/\tau}) \leq (1-p)\tau xe^{-p/\tau} = (1-p)ve^{-p/\tau} \to 0.$$

Therefore, the second term in (28) vanishes. For the remaining dominant term, let $z_1 = xe^{(1-p)/\tau} = \frac{v}{\tau}e^{(1-p)/\tau}$, then $z_1 \to \infty$ when $\tau \to 0$, and $\log z_1 = \frac{1-p}{\tau} + \log\frac{v}{\tau}$. In this case, our bound on the Lambert-$W$ function gives

$$\tau W(z_1) = (1-p) + o(1),$$

and thus

$$v = \tau x = \tau\left(pW(z_1) + o(1)\right) = p(1-p) + o(1) \implies \lambda = \tau p(1-p)(1 + o(1)).$$

$\square$

## B.2. Policy Ratio

We formally state the policy ratios of PMD-MEAN and PMD-PART in Equations (8) to (11) in the following proposition.

**Proposition B.1** (Binary-reward ratios for small $\tau$). *Assume $r(y) \in \{0, 1\}$ and let $p = \mathbb{E}_{\pi_t}[r(y)] \in (0, 1)$. Consider the ratios $\rho(y) := \pi_{t+1}(y)/\pi_t(y)$.*

**PMD-MEAN.** *As $\tau \to 0$, for any $y$ with $r(y) = 1$,*

$$\rho_+^{\text{mean}}(y) = \frac{1}{p} - \frac{1-p}{p}e^{-p/\tau}(1 + o(1)),$$

*and for any $y$ with $r(y) = 0$,*

$$\rho_-^{\text{mean}}(y) = e^{-p/\tau}(1 + o(1)).$$

**PMD-PART.** *For any $\tau > 0$, the partition update satisfies*

$$\rho_+^{\text{part}}(y) = \frac{1}{p + (1-p)e^{-1/\tau}}, \quad \rho_-^{\text{part}}(y) = \frac{e^{-1/\tau}}{p + (1-p)e^{-1/\tau}},$$

*and in particular as $\tau \to 0$,*

$$\rho_+^{\text{part}}(y) = \frac{1}{p} - \frac{1-p}{p^2}e^{-1/\tau} + O(e^{-2/\tau}), \quad \rho_-^{\text{part}}(y) = \frac{1}{p}e^{-1/\tau} + O(e^{-2/\tau}).$$

*Proof of Proposition B.1.* Throughout, let $x := \lambda/\tau^2$. For PMD-MEAN, start from the Lambert-$W$ form (5) and use $e^{-W(z)} = W(z)/z$ to rewrite the ratio as

$$\frac{\pi_{t+1}^{\text{mean}}(y)}{\pi_t(y)} = \exp\left(\frac{\Delta_y}{\tau} - W(xe^{\Delta_y/\tau})\right) = \frac{1}{x}W\left(xe^{\Delta_y/\tau}\right). \tag{29}$$

In the binary case, $\Delta = 1 - p$ when $r = 1$ and $\Delta = -p$ when $r = 0$, so defining

$$\rho_+^{\text{mean}} := \frac{1}{x}W\left(xe^{(1-p)/\tau}\right), \quad \rho_-^{\text{mean}} := \frac{1}{x}W\left(xe^{-p/\tau}\right),$$

we have the normalization identity

$$1 = \sum_y \pi_{t+1}^{\text{mean}}(y) = p\rho_+^{\text{mean}} + (1 - p)\rho_-^{\text{mean}}. \tag{30}$$

By (7), $\lambda \sim \tau p(1 - p)$ as $\tau \to 0$, hence

$$x = \frac{\lambda}{\tau^2} \sim \frac{p(1 - p)}{\tau} = \Theta\left(\frac{1}{\tau}\right).$$

Therefore $xe^{-p/\tau} \to 0$ as $\tau \to 0$. Using the Taylor expansion $W(z) = z + O(z^2)$ as $z \to 0$, we obtain

$$W(xe^{-p/\tau}) = xe^{-p/\tau}(1 + o(1)),$$

and plugging into (29) yields

$$\rho_-^{\text{mean}} = \frac{1}{x}W(xe^{-p/\tau}) = e^{-p/\tau}(1 + o(1)).$$

For $\rho_+^{\text{mean}}$, substituting the above into (30) gives

$$\rho_+^{\text{mean}} = \frac{1 - (1 - p)\rho_-^{\text{mean}}}{p} = \frac{1}{p} - \frac{1 - p}{p}e^{-p/\tau}(1 + o(1)).$$

This proves the PMD-MEAN claims.

For PMD-PART, the update is explicit:

$$\pi_{t+1}^{\text{part}}(y) = \frac{\pi_t(y)e^{r(y)/\tau}}{pe^{1/\tau} + (1 - p)}.$$

Hence

$$\rho_+^{\text{part}} = \frac{e^{1/\tau}}{pe^{1/\tau} + (1 - p)} = \frac{1}{p + (1 - p)e^{-1/\tau}}, \quad \rho_-^{\text{part}} = \frac{1}{pe^{1/\tau} + (1 - p)} = \frac{e^{-1/\tau}}{p + (1 - p)e^{-1/\tau}}.$$

Expanding $(1 + u)^{-1} = 1 - u + O(u^2)$ with $u = \frac{1-p}{p}e^{-1/\tau}$ yields

$$\rho_+^{\text{part}} = \frac{1}{p} - \frac{1 - p}{p^2}e^{-1/\tau} + O(e^{-2/\tau}), \quad \rho_-^{\text{part}} = \frac{1}{p}e^{-1/\tau} + O(e^{-2/\tau}).$$

$\square$

## B.3. Proof of Proposition 3.3

*Proof of Proposition 3.3.* Fix a state and omit $x$. Let $u(y) := \log\frac{\pi(y)}{\pi_t(y)}$. Then the simplex constraint $\sum_y \pi(y) = 1$ is equivalent to the single equality constraint

$$\mathbb{E}_{y \sim \pi_t}\left[e^{u(y)}\right] = 1. \tag{31}$$

Hence the mixed subproblem (12) is equivalent to

$$\max_{u:\, \mathbb{E}_{\pi_t}[e^u]=1} \mathbb{E}_{\pi_t}\left[e^u r - \tau e^u u - \frac{\lambda}{2\tau}(e^u - 1)^2\right].$$

Introduce a Lagrange multiplier $\nu \in \mathbb{R}$ for the constraint (31) and define

$$\mathcal{L}(u,\nu) := \mathbb{E}_{\pi_t}\left[e^u r - \tau e^u u - \frac{\lambda}{2\tau}(e^u - 1)^2\right] + \nu\big(\mathbb{E}_{\pi_t}[e^u] - 1\big).$$

Stationarity w.r.t. $u(y)$ gives, for all $y$,

$$0 = \pi_t(y)e^{u(y)}\Big(r(y) - \tau(u(y) + 1) - \frac{\lambda}{\tau}(e^{u(y)} - 1) + \nu\Big).$$

Dividing $\pi_t(y)e^{u(y)} > 0$ and rearranging the terms give

$$u(y) - \frac{r(y)}{\tau} + \frac{\lambda}{\tau^2}e^{u(y)} = c, \quad c := \frac{\nu + \lambda/\tau - \tau}{\tau}, \tag{32}$$

where $c$ is a constant independent of $y$.

Now specialize $u$ to the PMD-MEAN log-ratio $u(y) = \log\frac{\pi_{t+1}(y)}{\pi_t(y)}$ from Theorem 3.1. Its KKT conditions in Section B.1, with the same normalization multiplier $\lambda$, are

$$u(y) - \frac{\Delta_y}{\tau} + \frac{\lambda}{\tau^2}e^{u(y)} = 0, \quad \mathbb{E}_{\pi_t}[e^{u(y)}] = 1. \tag{33}$$

Let $\mu := \mathbb{E}_{\pi_t}[r]$, so that $\Delta_y = r(y) - \mu$. Then (33) implies

$$u(y) - \frac{r(y)}{\tau} + \frac{\lambda}{\tau^2}e^{u(y)} = -\frac{\mu}{\tau}.$$

This is exactly (32) with $\nu = \tau - \lambda/\tau - \mu$, since then $c = (\nu + \lambda/\tau - \tau)/\tau = -\mu/\tau$. Hence the PMD-MEAN solution satisfies the KKT conditions of the mixed subproblem (12) with the same $\lambda$.

Finally, the mixed objective in (12) is strictly concave in $\pi$ on the simplex: the reward term is linear, $-\tau\mathrm{KL}(\pi \parallel \pi_t)$ is strictly concave for $\tau > 0$, and $-(\lambda/(2\tau))\chi^2(\pi \parallel \pi_t)$ is concave for $\lambda \geq 0$. Therefore the KKT point is the unique global maximizer, so $\pi_{t+1}$ solves (12). $\qquad\square$

## B.4. General Bounded Rewards

The exact solution in Theorem 3.1 and the mixed-regularization view in Proposition 3.3 hold for any bounded reward through the advantage $\Delta_y = r(y) - \mathbb{E}_{\pi_t}[r(y)]$. The binary specialization is used only to express the log-ratio bounds and target-estimation terms through the single pass rate $p_t$. For general bounded rewards, analogous statements depend on the distribution of $\Delta_y$ under $\pi_t$, such as exponential moments appearing in the normalization constant and reward-conditional mass in high advantage regions. Thus the qualitative mechanism remains the same, while the closed-form constants are less interpretable.

## B.5. Length-Normalized PMD-MEAN

The implementation in Section A uses the sequence-level loss weighted by $\tau/|y|$. For a fixed prompt, write $u(y) = \log\frac{\pi(y)}{\pi_t(y)}$ and let $w(y) > 0$ denote an action-dependent weight. Ignoring global constants that do not change the minimizer, the weighted population objective can be written in the same scaling as the proof of Theorem 3.1:

$$\min_{u:\mathbb{E}_{\pi_t}[e^{u(y)}]=1} \frac{1}{2}\mathbb{E}_{\pi_t}\left[w(y)\left(\Delta_y - \tau u(y)\right)^2\right]. \tag{34}$$

Following the derivations as in Section B.1, the KKT conditions give

$$w(y)\left(u(y) - \frac{\Delta_y}{\tau}\right) + \frac{\lambda}{\tau^2}e^{u(y)} = 0, \tag{35}$$

and hence

$$u(y) = \frac{\Delta_y}{\tau} - W\left(\frac{\lambda}{\tau^2 w(y)} \exp\left(\frac{\Delta_y}{\tau}\right)\right), \quad \pi(y) = \pi_t(y) \exp(u(y)), \tag{36}$$

where $\lambda \geq 0$ is chosen so that $\mathbb{E}_{\pi_t}[e^{u(y)}] = 1$. For the implemented length normalization, $w(y)$ is proportional to $1/|y|$.

Equivalently, the same first-order condition is obtained from the KL-regularized subproblem

$$\pi_w = \underset{\pi \in \Delta(\mathcal{Y})}{\operatorname{argmax}} \mathbb{E}_{y \sim \pi}[r(y)] - \tau \mathrm{KL}(\pi \| \pi_t) - \frac{\lambda}{2\tau} \mathbb{E}_{y \sim \pi_t}\left[\frac{1}{w(y)} \left(\frac{\pi(y)}{\pi_t(y)}\right)^2\right], \tag{37}$$

where the same adaptive multiplier $\lambda$ appears in (36). The last term is the additional regularizer analogous to the last term in (12). It penalizes squared probability ratios, with action-dependent strength $1/w(y)$. For the implemented choice $w(y) = 1/|y|$, longer sequences receive a larger penalty on large sequence-level probability-ratio spikes. When $w(y)$ is constant, the last term is equivalent on the simplex to a constant multiple of $\chi^2(\pi \| \pi_t)$, recovering Proposition 3.3 up to constants. The above derivation shows that the same Lambert-$W$ normalization mechanism persists in the implemented loss in Section A, with an additional length reweighting that introduces stronger regularization on long responses.

## C. Missing Proofs in Section 4.1

We first state a simple self-bounding lemma from Bernstein's inequality.

**Lemma C.1** (Bernstein with self-bounding variance)**.** *Let $Z_1, \ldots, Z_n$ be i.i.d. random variables such that $\mathbb{E}[Z_i] = \mu \geq 0$, $|Z_i| \leq R$, and $\mathbb{E}[Z_i^2] \leq v\mu$ for some $v > 0$. Then for any $\delta \in (0,1)$, with probability at least $1 - \delta$,*

$$\mu \leq 2 \cdot \frac{1}{n} \sum_{i=1}^{n} Z_i + \frac{(2v + \frac{4}{3}R)\log(1/\delta)}{n}. \tag{38}$$

*Proof.* By Bernstein's inequality for bounded variables, with probability at least $1 - \delta$,

$$\mu \leq \frac{1}{n} \sum_{i=1}^{n} Z_i + \sqrt{\frac{2\mathrm{Var}(Z_i)\log(1/\delta)}{n}} + \frac{2R\log(1/\delta)}{3n}.$$

Using $\mathrm{Var}(Z_i) \leq \mathbb{E}[Z_i^2] \leq v\mu$ yields

$$\mu \leq \frac{1}{n} \sum_{i=1}^{n} Z_i + \sqrt{\frac{2v\mu\log(1/\delta)}{n}} + \frac{2R\log(1/\delta)}{3n}.$$

Apply $\sqrt{ab} \leq \frac{1}{2}a + \frac{1}{2}b$ with $a = \mu$ and $b = \frac{2v\log(1/\delta)}{n}$:

$$\sqrt{\frac{2v\mu\log(1/\delta)}{n}} \leq \frac{\mu}{2} + \frac{v\log(1/\delta)}{n}.$$

Substitute and rearrange to obtain (38). $\qquad\square$

### C.1. Proof of Lemma 4.5

*Proof of Lemma 4.5.* Fix iteration $t$. For each $\pi \in \Pi$, define the residual

$$f_\pi(y) := s_\pi(y) - s^\star(y).$$

By Assumption 4.1, $s^\star = s_{\pi_{t+1}^\star}$ for some $\pi_{t+1}^\star \in \Pi$. Hence by Assumption 4.3, for all $\pi \in \Pi$ and $y \in \mathcal{Y}$,

$$|f_\pi(y)| \leq |s_\pi(y)| + |s^\star(y)| \leq 2B =: M.$$

Define the clean empirical loss

$$\widehat{\mathcal{L}}_t^{\text{clean}}(\pi) := \frac{1}{2n}\sum_{i=1}^{n} f_\pi(y_i)^2 = \frac{1}{n}\sum_{i=1}^{n} X_i(\pi), \quad X_i(\pi) := \frac{1}{2}f_\pi(y_i)^2.$$

Then $0 \le X_i(\pi) \le \frac{1}{2}M^2$ and $\mathbb{E}[X_i(\pi)] = \mathcal{L}_t(\pi)$. Applying Lemma C.1 with $v = R = \frac{1}{2}M^2$ and a union bound over $\pi \in \Pi$, we get that with probability at least $1 - \delta$, for all $\pi \in \Pi$,

$$\mathcal{L}_t(\pi) \le 2\widehat{\mathcal{L}}_t^{\text{clean}}(\pi) + \frac{5M^2\log(|\Pi|/\delta)}{3n} < 2\widehat{\mathcal{L}}_t^{\text{clean}}(\pi) + \frac{5M^2\log(2|\Pi|/\delta)}{3n}. \tag{39}$$

Next, relate $\widehat{\mathcal{L}}_t^{\text{clean}}(\widehat{\pi}_{t+1})$ to the target mismatch $\overline{\Delta^2}$. For each $i$, write $\Delta_i = \widetilde{s}_{-i}^\star(y_i) - s^\star(y_i)$ so that

$$s_{\widehat{\pi}_{t+1}}(y_i) - \widetilde{s}_{-i}^\star(y_i) = f_{\widehat{\pi}_{t+1}}(y_i) - \Delta_i.$$

Using Assumptions 4.1 and 4.2, we get

$$\widehat{\mathcal{L}}_t(\widehat{\pi}_{t+1}) \le \inf_{\pi\in\Pi}\widehat{\mathcal{L}}_t(\pi) + \epsilon_{\text{opt}} \le \widehat{\mathcal{L}}_t(\pi_{t+1}^\star) + \epsilon_{\text{opt}} = \frac{1}{2n}\sum_{i=1}^{n}\Delta_i^2 + \epsilon_{\text{opt}} = \frac{1}{2}\overline{\Delta^2} + \epsilon_{\text{opt}}.$$

On the other hand, the pointwise inequality $(u - v)^2 \ge \frac{1}{2}u^2 - v^2$ implies

$$\widehat{\mathcal{L}}_t(\widehat{\pi}_{t+1}) = \frac{1}{2n}\sum_{i=1}^{n}\left(f_{\widehat{\pi}_{t+1}}(y_i) - \Delta_i\right)^2 \ge \frac{1}{4n}\sum_{i=1}^{n} f_{\widehat{\pi}_{t+1}}(y_i)^2 - \frac{1}{2n}\sum_{i=1}^{n}\Delta_i^2 = \frac{1}{2}\widehat{\mathcal{L}}_t^{\text{clean}}(\widehat{\pi}_{t+1}) - \frac{1}{2}\overline{\Delta^2}.$$

Combining the last two displays yields

$$\widehat{\mathcal{L}}_t^{\text{clean}}(\widehat{\pi}_{t+1}) \le 2\overline{\Delta^2} + 2\epsilon_{\text{opt}}.$$

Finally, apply (39) at $\pi = \widehat{\pi}_{t+1}$ and substitute the above bound, we get

$$\mathcal{L}_t(\widehat{\pi}_{t+1}) \le 2\widehat{\mathcal{L}}_t^{\text{clean}}(\widehat{\pi}_{t+1}) + \frac{5M^2\log(2|\Pi|/\delta)}{3n} \le 4\overline{\Delta^2} + 4\epsilon_{\text{opt}} + \frac{5M^2\log(2|\Pi|/\delta)}{3n},$$

which proves (20) as $M = 2B$. $\qquad\qquad\square$

### C.2. Proof of Theorem 4.6

*Proof of Theorem 4.6.* Since $r \in [0, 1]$, for any two policies $p, q$ we have $|J(p) - J(q)| \le \text{TV}(p, q)$. Moreover,

$$\begin{aligned}
\text{TV}(\pi_{t+1}, \pi_{t+1}^\star) &= \frac{1}{2}\mathbb{E}_{y\sim\pi_t}\left[\left|e^{s_{\pi_{t+1}}(y)} - e^{s^\star(y)}\right|\right] \\
&\le \frac{1}{2}\mathbb{E}_{y\sim\pi_t}\left[\max\{e^{s_{\pi_{t+1}}(y)}, e^{s^\star(y)}\}\cdot\left|s_{\pi_{t+1}}(y) - s^\star(y)\right|\right] \\
&\le \frac{1}{2}\left(\sqrt{\mathbb{E}_{\pi_t}\left[e^{2s_{\pi_{t+1}}(y)}\right]} + \sqrt{\mathbb{E}_{\pi_t}\left[e^{2s^\star(y)}\right]}\right)\cdot\sqrt{\mathbb{E}_{\pi_t}\left[(s_{\pi_{t+1}}(y) - s^\star(y))^2\right]} \\
&= \frac{1}{2}\left(\sqrt{1 + \chi^2(\pi_{t+1}\|\pi_t)} + \sqrt{1 + \chi^2(\pi_{t+1}^\star\|\pi_t)}\right)\cdot\sqrt{2\mathcal{L}_t(\pi_{t+1})},
\end{aligned}$$

where we used $\left|e^a - e^b\right| \le \max\{e^a, e^b\}|a - b|$ and Cauchy–Schwarz. Since $s_\pi \le B_+$ and $\mathbb{E}_{\pi_t}[e^{s_\pi}] = 1$, we have $\mathbb{E}_{\pi_t}[e^{2s_\pi}] \le e^{B_+}$, hence the prefactor is at most $e^{B_+/2}$ up to constants. Combining the assumption on the ideal convergence rate (21) and the bound on $\mathcal{L}_t(\pi_{t+1})$ in Lemma 4.5, we obtain the result (22). $\qquad\square$

## C.3. Proof of Proposition 4.7

*Proof of Proposition 4.7.* By (10) in Section 3.1, for $r(y) = 0$,

$$\frac{\pi_{t+1}^{\mathrm{mean}}(y)}{\pi_t(y)} = \exp\left(-\frac{p_t}{\tau}\right)(1 + o(1)).$$

Therefore, the total probability mass on the negative set contracts as

$$1 - p_{t+1}^\star = \mathbb{E}_{y \sim \pi_t}\left[\frac{\pi_{t+1}^\star(y)}{\pi_t(y)} \cdot \mathbf{1}\{r(y) = 0\}\right]$$
$$= (1 - p_t)\exp\left(-\frac{p_t}{\tau}\right)(1 + o(1)),$$

which yields (23). □

## C.4. Proof of Proposition 4.8

*Proof of Proposition 4.8.* Under (2), the total probability mass on $r = 1$ is

$$p_{t+1}^\star = \frac{p_t e^{1/\tau}}{p_t e^{1/\tau} + (1 - p_t)},$$

which implies $1 - p_{t+1}^\star = \frac{1-p_t}{p_t e^{1/\tau}+(1-p_t)}$. This is exactly (21) with (24). □

## C.5. Proof of Proposition 4.11

**Lemma C.2** (LOO mean concentration). *Let $U_1, \ldots, U_n$ be i.i.d. Bernoulli random variables with mean $p$ and define $p_{-i} = \frac{1}{n-1}\sum_{j \neq i} U_j$. Then for any $\delta \in (0, 1)$, with probability at least $1 - \delta$,*

$$\max_{i \in [n]}|p_{-i} - p| \leq \sqrt{\frac{2p(1-p)\log(4n/\delta)}{n-1}} + \frac{2\log(4n/\delta)}{3(n-1)}.$$

*Proof.* For any $i$, by Bernstein's inequality for $\sum_{j \neq i}(U_j - p)$, with probability at least $1 - \delta/(2n)$,

$$|p_{-i} - p| \leq \sqrt{\frac{2p(1-p)\log(\frac{2n}{\delta})}{n-1}} + \frac{2\log(\frac{2n}{\delta})}{3(n-1)}.$$

Applying a union bound over $i \in [n]$ and both signs gives the stated bound. □

*Proof of Proposition 4.11.* Applying Lemma C.2 to $U_i = r(y_i)$, we obtain $\max_i |p_{-i} - p_t| \leq \varepsilon_n(p_t, \delta)$ with probability at least $1 - \delta$.

**PMD-PART.** For PMD-PART, write $a = e^{1/\tau} - 1$ and $Z_t = 1 + ap_t$. Since $r(y_i) \in \{0, 1\}$, we have

$$\Delta_i = \log(1 + ap_t) - \log(1 + ap_{-i}) = \frac{a}{1 + a\xi_i}(p_t - p_{-i})$$

for some $\xi_i$ between $p_t$ and $p_{-i}$ by the mean value theorem. Therefore,

$$|\Delta_i| \leq \frac{a}{1 + a(p_t - |p_{-i} - p_t|)_+}\,|p_{-i} - p_t|.$$

Meanwhile, there is always $\tau \log(1 + ap) \in [0, 1]$, and thus $|\Delta_i| \leq \frac{1}{\tau}$. In the event that $\max_{i \in [n]}|p_{-i} - p_t| \leq \varepsilon_n(p_t, \delta)$, combining the inequalities, squaring and averaging over $i$ yields (26).

**PMD-MEAN.** Recall that $\widetilde{s}_{-i}^\star(y_i)$ is defined in (17) and $s^\star(y) = \log\frac{\pi_{t+1}^{\mathrm{mean}}(y)}{\pi_t(y)}$. By (5), we have

$$s^\star(y) = \frac{r(y) - p_t}{\tau} - W\left(\frac{\lambda}{\tau^2}\exp\left(\frac{r(y) - p_t}{\tau}\right)\right).$$

Therefore, for each $i \in [n]$,

$$\Delta_i = \widetilde{s}^\star_{-i}(y_i) - s^\star(y_i) = \frac{p_t - p_{-i}}{\tau} + W\Big(\frac{\lambda}{\tau^2} \exp\Big(\frac{r(y_i) - p_t}{\tau}\Big)\Big).$$

Using $(a + b)^2 \le 2a^2 + 2b^2$ and averaging over $i$ gives

$$\overline{\Delta^2} \le \frac{2 \max_i |p_{-i} - p_t|^2}{\tau^2} + \frac{2}{n} \sum_{i=1}^n W\Big(\frac{\lambda}{\tau^2} \exp\Big(\frac{r(y_i) - p_t}{\tau}\Big)\Big)^2.$$

On the event $\max_i |p_{-i} - p_t| \le \varepsilon_n(p_t, \delta)$, the first term is bounded by $\frac{2\varepsilon_n(p_t, \delta)^2}{\tau^2}$. For the second term, since $r(y_i) \in \{0, 1\}$ it takes only two values:

$$w_+ := W\Big(\frac{\lambda}{\tau^2} \exp\Big(\frac{1 - p_t}{\tau}\Big)\Big), \quad w_- := W\Big(\frac{\lambda}{\tau^2} \exp\Big(-\frac{p_t}{\tau}\Big)\Big).$$

Writing $\widehat{p}_t := \frac{1}{n} \sum_{i=1}^n r(y_i)$, we have

$$\frac{1}{n} \sum_{i=1}^n W\Big(\frac{\lambda}{\tau^2} \exp\Big(\frac{r(y_i) - p_t}{\tau}\Big)\Big)^2 = \widehat{p}_t w_+^2 + (1 - \widehat{p}_t) w_-^2.$$

By the asymptotics used in the proof of Theorem 3.1 in Section B.1, as $\tau \to 0$ with fixed $p_t \in (0, 1)$ we have $\tau w_+ = (1 - p_t) + o(1)$ and $\tau w_- = o(1)$, and hence for sufficiently small $\tau$,

$$w_+^2 \lesssim \frac{(1 - p_t)^2}{\tau^2}, \quad w_-^2 = o\Big(\frac{1}{\tau^2}\Big).$$

Since $\max_i |p_{-i} - p_t| \le \varepsilon_n(p_t, \delta)$, we have

$$|\widehat{p}_t - p_t| = \Big|\frac{1}{n} \sum_{i=1}^n (p_{-i} - p_t)\Big| \le \frac{1}{n} \sum_{i=1}^n |p_{-i} - p_t| \le \varepsilon_n(p_t, \delta).$$

Combining the above bounds yields

$$\frac{1}{n} \sum_{i=1}^n W\Big(\frac{\lambda}{\tau^2} \exp\Big(\frac{r(y_i) - p_t}{\tau}\Big)\Big)^2 \lesssim \frac{p_t(1 - p_t)^2}{\tau^2} (1 + \varepsilon_n(p_t, \delta)).$$

Substituting back proves (25). $\qquad\qquad\qquad\qquad\qquad\qquad\qquad\qquad\qquad\qquad\qquad\qquad\qquad\qquad\square$

## D. Refined Analysis for PMD-MEAN

To refine the analysis of PMD-MEAN, we first connect the population squared loss $\mathcal{L}_t$ in (15) with ideal target $s^\star(y) = \log \frac{\pi_{t+1}^{\mathrm{mean}}(y)}{\pi_t(y)}$ with the population objective of PMD-MEAN in (4).

**Lemma D.1** (Connection of losses for PMD-MEAN). *Fix a global step $t$ and write $\Delta_y := r(y) - \mathbb{E}_{y' \sim \pi_t}[r(y')]$. Define the PMD-MEAN population objective, i.e., bandit specialization of (4):*

$$\mathcal{L}_t^{\mathrm{mean}}(\pi) := \frac{1}{2} \mathbb{E}_{y \sim \pi_t}\Big[\Big(s_\pi(y) - \frac{\Delta_y}{\tau}\Big)^2\Big]. \tag{40}$$

*Let $\pi_{t+1}^\star$ be the ideal PMD-MEAN update and $s^\star = s_{\pi_{t+1}^\star}$. Then for any $\pi \in \Pi$,*

$$\mathcal{L}_t^{\mathrm{mean}}(\pi) - \mathcal{L}_t^{\mathrm{mean}}(\pi_{t+1}^\star) = \mathcal{L}_t(\pi) + \frac{\lambda}{\tau^2} \mathrm{KL}(\pi_{t+1}^\star \| \pi) \ge \mathcal{L}_t(\pi), \tag{41}$$

*where $\lambda \ge 0$ is the KKT dual multiplier in (33).*

Using Lemma D.1, we can refine the ERM analysis for PMD-MEAN and eliminate the error floor in target estimation error.

**Lemma D.2** (Refined ERM for PMD-MEAN)**.** *Suppose Assumptions 4.1 to 4.4 hold, $r(y) \in \{0,1\}$ and define $p_t :=$ $\mathbb{E}_{y \sim \pi_t}[r(y)]$. Let $\varepsilon_n(p_t, \delta)$ be as in Proposition 4.11. Then for any $\delta \in (0,1)$, with probability at least $1 - \delta$,*

$$\mathcal{L}_t(\widehat{\pi}_{t+1}) \lesssim \frac{(B + \frac{1}{\tau})^2 \log(|\Pi| / \delta)}{n} + \epsilon_{\text{opt}} + \frac{\varepsilon_n(p_t, \delta)}{\tau}\left(B + \frac{p_t}{\tau}\right) + \frac{\varepsilon_n(p_t, \delta)^2}{\tau^2}. \tag{42}$$

By Proposition 4.9, for fixed $p_t \in (0,1)$ and sufficiently small $\tau > 0$, we may take $B \lesssim p_t/\tau$ for PMD-MEAN. Substituting this in (42) gives Lemma 4.12 in Section 4.

### D.1. Proof of Lemma D.1

*Proof of Lemma D.1.* Denote $g(y) := \Delta_y/\tau$ for brevity. For any $\pi \in \Pi$, we have

$$\begin{aligned}
\mathcal{L}_t^{\text{mean}}(\pi) - \mathcal{L}_t^{\text{mean}}(\pi_{t+1}^\star) &= \frac{1}{2}\mathbb{E}_{\pi_t}\left[(s_\pi - g)^2 - (s^\star - g)^2\right] \\
&= \frac{1}{2}\mathbb{E}_{\pi_t}\left[(s_\pi - s^\star)^2\right] + \mathbb{E}_{\pi_t}\left[(s_\pi - s^\star)(s^\star - g)\right] \\
&= \mathcal{L}_t(\pi) + \mathbb{E}_{\pi_t}\left[(s_\pi - s^\star)(s^\star - g)\right].
\end{aligned}$$

By the KKT conditions (33) with $u = s^\star$,

$$s^\star(y) - g(y) = -\frac{\lambda}{\tau^2}e^{s^\star(y)}.$$

Combining the identities and $e^{s^\star(y)} = \pi_{t+1}^\star(y)/\pi_t(y)$, we get

$$\begin{aligned}
\mathbb{E}_{\pi_t}\left[(s_\pi - s^\star)(s^\star - g)\right] &= -\frac{\lambda}{\tau^2}\mathbb{E}_{\pi_t}\left[(s_\pi - s^\star)e^{s^\star}\right] \\
&= -\frac{\lambda}{\tau^2}\mathbb{E}_{y \sim \pi_{t+1}^\star}\left[\log\frac{\pi(y)}{\pi_{t+1}^\star(y)}\right] \\
&= \frac{\lambda}{\tau^2}\text{KL}(\pi_{t+1}^\star \| \pi) \geq 0,
\end{aligned}$$

which proves (41). $\qquad\square$

### D.2. Proof of Lemma D.2

*Proof of Lemma D.2.* Recall the PMD-MEAN leave-one-out target (17):

$$\widetilde{s}_{-i}^\star(y_i) = \frac{1}{\tau}\left(r(y_i) - p_{-i}\right), \quad p_{-i} := \frac{1}{n-1}\sum_{j \neq i} r(y_j).$$

Also recall $p_t = \mathbb{E}_{y \sim \pi_t}[r(y)]$ and $\Delta_{y_i} = r(y_i) - p_t$.

We first decompose the empirical targets as

$$\widetilde{s}_{-i}^\star(y_i) = \frac{\Delta_{y_i}}{\tau} + \Delta_i^{\text{loo}}, \quad \Delta_i^{\text{loo}} := \frac{p_t - p_{-i}}{\tau}. \tag{43}$$

Define the empirical loss with the population baseline target:

$$\widehat{\mathcal{L}}_t^{\text{mean}}(\pi) := \frac{1}{2n}\sum_{i=1}^n\left(s_\pi(y_i) - \frac{\Delta_{y_i}}{\tau}\right)^2. \tag{44}$$

By (43), for each $\pi \in \Pi$,

$$\begin{aligned}
\widehat{\mathcal{L}}_t(\pi) - \widehat{\mathcal{L}}_t^{\text{mean}}(\pi) &= \frac{1}{2n}\sum_{i=1}^n\left[(a_i - \Delta_i^{\text{loo}})^2 - a_i^2\right] \\
&= \frac{1}{2n}\sum_{i=1}^n\left[(\Delta_i^{\text{loo}})^2 - 2a_i \cdot \Delta_i^{\text{loo}}\right],
\end{aligned}$$

where $a_i := s_\pi(y_i) - \frac{\Delta_{y_i}}{\tau}$. Thus we have

$$\left|\widehat{\mathcal{L}}_t(\pi) - \widehat{\mathcal{L}}_t^{\text{mean}}(\pi)\right| \leq \frac{1}{n}\sum_{i=1}^{n}|a_i| \cdot \left|\Delta_i^{\text{loo}}\right| + \frac{1}{2n}\sum_{i=1}^{n}(\Delta_i^{\text{loo}})^2. \tag{45}$$

Let $\mathcal{E}$ be the event from Proposition 4.11 that $\max_i |p_{-i} - p_t| \leq \varepsilon_n(p_t, \delta/2)$. On $\mathcal{E}$ we have $\left|\Delta_i^{\text{loo}}\right| \leq \varepsilon_n(p_t, \delta/2)/\tau$, hence

$$\sup_{\pi \in \Pi}\left|\widehat{\mathcal{L}}_t(\pi) - \widehat{\mathcal{L}}_t^{\text{mean}}(\pi)\right| \leq \frac{\varepsilon_n}{\tau} \cdot \sup_{\pi \in \Pi}\frac{1}{n}\sum_{i=1}^{n}|a_i| + \frac{\varepsilon_n^2}{2\tau^2}, \tag{46}$$

where we write $\varepsilon_n := \varepsilon_n(p_t, \delta/2)$.

We now bound the $\frac{1}{n}\sum_{i=1}^{n}|a_i|$ term. By Assumption 4.3, $|s_\pi(y)| \leq B$ for all $\pi \in \Pi, y \in \mathcal{Y}$. Also, since $r(y_i) \in \{0, 1\}$,

$$|\Delta_{y_i}| = |r(y_i) - p_t| = \begin{cases} p_t, & r(y_i) = 0, \\ 1 - p_t, & r(y_i) = 1. \end{cases}$$

Thus for any $\pi$ and any $i$,

$$|a_i| = \left|s_\pi(y_i) - \frac{\Delta_{y_i}}{\tau}\right| \leq |s_\pi(y_i)| + \frac{|\Delta_{y_i}|}{\tau} \leq B + \frac{|r(y_i) - p_t|}{\tau}.$$

Averaging over $i$ yields

$$\frac{1}{n}\sum_{i=1}^{n}|a_i| \leq B + \frac{1}{\tau} \cdot \frac{1}{n}\sum_{i=1}^{n}|r(y_i) - p_t|. \tag{47}$$

Let $\bar{r} := \frac{1}{n}\sum_{i=1}^{n}r(y_i)$. For binary rewards,

$$\frac{1}{n}\sum_{i=1}^{n}|r(y_i) - p_t| = p_t + (1 - 2p_t)\bar{r} \leq 2p_t + |\bar{r} - p_t|.$$

Moreover, for any fixed $i$,

$$\bar{r} = \frac{(n-1)p_{-i} + r(y_i)}{n},$$

hence

$$|\bar{r} - p_t| \leq \frac{n-1}{n}|p_{-i} - p_t| + \frac{1}{n}|r(y_i) - p_t| \leq \max_j |p_{-j} - p_t| + \frac{1}{n}.$$

On $\mathcal{E}$ this gives $|\bar{r} - p_t| \leq \varepsilon_n + \frac{1}{n}$, and therefore

$$\frac{1}{n}\sum_{i=1}^{n}|r(y_i) - p_t| \leq 2p_t + \varepsilon_n + \frac{1}{n}. \tag{48}$$

Combining (47) and (48) and substituting into (46) yields

$$\sup_{\pi \in \Pi}\left|\widehat{\mathcal{L}}_t(\pi) - \widehat{\mathcal{L}}_t^{\text{mean}}(\pi)\right| \lesssim \frac{\varepsilon_n}{\tau}\left(B + \frac{p_t}{\tau} + \frac{\varepsilon_n}{\tau} + \frac{1}{n\tau}\right)$$

$$\lesssim \frac{\varepsilon_n}{\tau}\left(B + \frac{p_t}{\tau}\right) + \frac{\varepsilon_n^2}{\tau^2}. \tag{49}$$

We now bound the population excess risk of PMD-MEAN. Recall that

$$\mathcal{L}_t^{\text{mean}}(\pi) := \frac{1}{2}\mathbb{E}_{y\sim\pi_t}\left[\left(s_\pi(y) - \frac{\Delta_y}{\tau}\right)^2\right],$$

$$\widehat{\mathcal{L}}_t^{\text{mean}}(\pi) := \frac{1}{2n}\sum_{i=1}^n\left(s_\pi(y_i) - \frac{\Delta_{y_i}}{\tau}\right)^2.$$

For $\pi \in \Pi$, define the pointwise loss

$$\ell_\pi(y) := \frac{1}{2}\left(s_\pi(y) - \frac{\Delta_y}{\tau}\right)^2, \quad \ell^\star(y) := \frac{1}{2}\left(s^\star(y) - \frac{\Delta_y}{\tau}\right)^2.$$

Let

$$Z_i(\pi) := \ell_\pi(y_i) - \ell^\star(y_i), \quad \mu(\pi) := \mathbb{E}_{\pi_t}[Z_i(\pi)] = \mathcal{L}_t^{\text{mean}}(\pi) - \mathcal{L}_t^{\text{mean}}(\pi_{t+1}^\star), \quad \widehat{\mu}(\pi) := \frac{1}{n}\sum_{i=1}^n Z_i(\pi).$$

By Assumption 4.3 and $|\Delta_y| \leq 1$, we have

$$0 \leq \ell_\pi(y) \leq \frac{1}{2}M_\tau^2,$$

where $M_\tau := B + \frac{1}{\tau}$, and hence $|Z_i(\pi)| \leq \frac{1}{2}M_\tau^2$. Moreover,

$$Z_i^2(\pi) = \frac{1}{4}(s^\star(y) - s_\pi(y))^2\left(s^\star(y) + s_\pi(y) - \frac{2\Delta_y}{\tau}\right)^2$$

$$\leq (s^\star(y) - s_\pi(y))^2 M_\tau^2,$$

thus we have

$$\mathbb{E}[Z_i^2(\pi)] \leq M_\tau^2\mathbb{E}[(s_\pi - s^\star)^2]$$

$$= 2M_\tau^2\mathcal{L}_t(\pi)$$

$$\leq 2M_\tau^2\mu(\pi),$$

where the last inequality comes from Lemma D.1. Apply Lemma C.1 with $v = 2M_\tau^2$ and $R = \frac{1}{2}M_\tau^2$, then with probability at least $1 - \delta'$, for a fixed $\pi$,

$$\mu(\pi) \leq 2\widehat{\mu}(\pi) + O\left(\frac{M_\tau^2\log(1/\delta')}{n}\right).$$

A union bound over $\Pi$ with $\delta' = \frac{\delta}{2|\Pi|}$ yields that with probability at least $1 - \frac{\delta}{2}$, for all $\pi \in \Pi$,

$$\mathcal{L}_t(\pi) \leq \mathcal{L}_t^{\text{mean}}(\pi) - \mathcal{L}_t^{\text{mean}}(\pi_{t+1}^\star) \lesssim \widehat{\mathcal{L}}_t^{\text{mean}}(\pi) - \widehat{\mathcal{L}}_t^{\text{mean}}(\pi_{t+1}^\star) + \frac{M_\tau^2\log(|\Pi|/\delta)}{n}. \tag{50}$$

We now bound the empirical excess risk.

$$\widehat{\mathcal{L}}_t^{\text{mean}}(\widehat{\pi}_{t+1}) - \widehat{\mathcal{L}}_t^{\text{mean}}(\pi_{t+1}^\star)$$

$$= \widehat{\mathcal{L}}_t^{\text{mean}}(\widehat{\pi}_{t+1}) - \widehat{\mathcal{L}}_t(\widehat{\pi}_{t+1}) + \underbrace{\widehat{\mathcal{L}}_t(\widehat{\pi}_{t+1}) - \widehat{\mathcal{L}}_t(\pi_{t+1}^\star)}_{\leq \epsilon_{\text{opt}}} + \widehat{\mathcal{L}}_t(\pi_{t+1}^\star) - \widehat{\mathcal{L}}_t^{\text{mean}}(\pi_{t+1}^\star)$$

$$\leq \epsilon_{\text{opt}} + 2\sup_{\pi\in\Pi}\left|\widehat{\mathcal{L}}_t(\pi) - \widehat{\mathcal{L}}_t^{\text{mean}}(\pi)\right|$$

$$\lesssim \epsilon_{\text{opt}} + \frac{\varepsilon_n}{\tau}\left(B + \frac{p_t}{\tau}\right) + \frac{\varepsilon_n^2}{\tau^2},$$

where the first inequality uses Assumptions 4.1 and 4.2 and the second inequality uses (49). Combining the above inequality with (50), we get

$$\mathcal{L}_t(\widehat{\pi}_{t+1}) \lesssim \frac{M_\tau^2\log(|\Pi|/\delta)}{n} + \epsilon_{\text{opt}} + \frac{\varepsilon_n}{\tau}\left(B + \frac{p_t}{\tau}\right) + \frac{\varepsilon_n^2}{\tau^2},$$

which is (42). $\qquad\square$

