# OpenReview forum: "Approximation of Log-Partition Function in Policy Mirror Descent Induces Implicit Regularization for LLM Post-Training"
_ICML.cc/2026/Conference — ICML 2026 regular_

### Official Review · Reviewer_XBBr · 2026-03-13

**Soundness:** 3
**Presentation:** 3
**Significance:** 3
**Originality:** 3
**Overall Recommendation:** 5
**Confidence:** 3

**Summary:**

This paper analyses a practical algorithm for LLM post training, called PMD-mean.
PMD-mean replaces the log partition function in the loss with the centered reward, which is not fully justified for finite inverse temperature $\tau$.
This paper theoretically shows that PMD-mean implicitly solving adaptive mixed KL-$\chi^2$ regularized problem, where the adaptivity comes from the fact that, since the policy is always forced to be a valid probability distribution, the regularization coefficient is determined implicitly so that the Lambert-W based solution is the valid probability distribution.
It was also shown by a series of analyses that,
though PMD-PART has a faster ideal convergence rate with large enough samples, PMD-MEAN can be more robust to statistical errors under limited samples.
Experimental results also show the robustness of PMD-MEAN that results in better performance in practice.

**Compliance With Llm Reviewing Policy:**

Affirmed.

**Final Justification:**

I keep my evaluation of `accept` since my concern, which was minor, was addressed.

**Key Questions For Authors:**

N/A

**Limitations:**

yes

**Strengths And Weaknesses:**

### Strengths
In general, I think this paper has solid contributions.
- This paper provide deeper understanding of the practical LLM post-training method.
  - The analyses show that PMD-mean is indeed a justified algorithm (Theorem 3.1 and Proposition 3.2).
  - Convergence properties are also examined, where the practical benefits are characterized (Theorem 4.6, Proposition 4.7 - 4.11).
  - Proofs seem technically sound, though I did not check the proofs rigorously.
- Experimental demonstrations of the robustness are also consistent with the theoretical results.

### Weaknesses
- (minor) Implications for better algorithmic design are weak.

---

> ### Author Rebuttal · Authors · 2026-03-31
>
> We thank the reviewer for the encouraging and careful review.
>
> > **[W1] Algorithmic Implications:** The implications for designing better, novel algorithms are somewhat weak.
>
> We agree that the current paper could say more about downstream algorithmic implications. Our main goal here is to characterize the sequence-level PMD-mean objective and its population solution, and explain why it behaves more conservatively and stably than PMD-part in the finite-sample regime. We believe this already offers useful design guidance. For example, it suggests that more adaptive forms of regularization may be beneficial in practice, especially when prompts differ substantially in difficulty or reward scale. In the revision, we will expand this discussion while keeping the paper focused on the core characterization results. We view the development of new algorithmic variants building on this analysis as an important direction for future work.

---

> > ### Author Rebuttal · Reviewer_XBBr · 2026-04-03
> >
> > The suggested expanded discussion is reasonable if concretely done.

---

### Official Review · Reviewer_avvY · 2026-03-13

**Soundness:** 3
**Presentation:** 3
**Significance:** 3
**Originality:** 3
**Overall Recommendation:** 4
**Confidence:** 2

**Summary:**

The paper studies the PMD-MEAN algorithm and shows its mean-reward approximation implicitly solves a mixed KL-$\chi^2$ regularized subproblem, explaining its empirical stability over the exact partition-function variant.

**Compliance With Llm Reviewing Policy:**

Affirmed.

**Final Justification:**

My major concerns have been addressed by the authors' rebuttal.

**Key Questions For Authors:**

While the authors mention the binary reward assumption is for simplification, can you elaborate on the difficulty of extending this to the general reward setting? Is this only for notational simplification of calculation, or does it face some structural challenges?Yes

**Limitations:**

yes

**Strengths And Weaknesses:**

Strengths:
This paper provides clean theoretical insight that neatly explains a practically deployed algorithm from a perspective of implicit regularization, and good separation analysis between PMD-MEAN and PMD-PART in the finite-sample regime.

Weakness:
Most theoretical insights are restricted to binary rewards. Such restriction prevents the insights from being applied to broader post-training settings with possibly continuous rewards.

---

> ### Author Rebuttal · Authors · 2026-03-31
>
> We thank the reviewer for the careful review and positive feedback.
>
> > **[W1&Q1] Binary Rewards Limit:** The theoretical insights are highly restricted to binary rewards, preventing immediate application to continuous reward settings. Is extending beyond binary rewards purely a matter of notational simplification, or are there fundamental structural challenges?
>
> Thanks for raising this point. Firstly, we clarify that our main results of **Theorem 3.1 and Proposition 3.2 do not require binary rewards**. Therefore, the mechanism of implicit regularization and the Lambert-W characterization of the optimal solution both hold for general rewards.
>
> On the other hand, the analysis of explicit policy ratio separations will be different in the general setting. In particular, for general continuous rewards, the tight upper bounds on the ratio $\frac{\pi_{t+1}}{\pi_t}$ in Eq. (8) and (9) do not hold. Instead, we have upper bounds in Eq. (13) and (14), which read $O(\frac{1}{\tau})$ for the implicit $\chi^2$ regularization, and $O(\exp(\frac{1}{\tau}))$ for the standard KL regularization, implying a **polynomial vs. exponential** separation between the two regularizers. In this case, the $B_+$ in Assumption 4.3 is much smaller for the mixed $\chi^2$ regularization, leading to a better bound in Eq. (22). Nevertheless, the bounds would be generally looser than the binary reward case, where the partition function is fully determined by the pass rate and hence enables exact characterizations.

---

> > ### Author Rebuttal · Reviewer_avvY · 2026-04-03
> >
> > Thank you for the rebuttal. My concerns are addressed.

---

### Official Review · Reviewer_kzUB · 2026-03-19

**Soundness:** 3
**Presentation:** 3
**Significance:** 4
**Originality:** 3
**Overall Recommendation:** 4
**Confidence:** 3

**Summary:**

This paper studies PMD-MEAN, a practical off-policy regression variant of KL-regularized policy mirror descent for LLM post-training, where the log-partition term is replaced by the rollout-policy mean reward and log-policy ratios are fit by squared regression. The main theoretical claim is that the population minimizer is not the standard Boltzmann PMD update, but a Lambert-W-form update that is exactly equivalent to solving a per-state mirror-descent problem with an adaptive mixed KL-χ² regularizer. The paper then gives a stylized small-τ, binary-reward convergence analysis arguing that this induces more conservative updates and greater robustness to finite-sample errors than direct partition fitting.

**Compliance With Llm Reviewing Policy:**

Affirmed.

**Final Justification:**

Most of my concerns were clarified by the authors.  I would like to keep positive Overall Recommendation and raise Significance score.

**Key Questions For Authors:**

1. **[Q1]** Does the exact Lambert-W / adaptive KL–χ² characterization still hold for the length-normalized loss actually used in Appendix A? If not, what is the correct population objective for the implemented algorithm?

2. **[Q2]** While the authors provide a good discussion connecting their mixed $\mathrm{KL}-\chi^2$ result to $\chi^2\mathrm{PO}$ (Huang et al., 2024) in Remarks 3.3/3.4, the positioning relative to the broader, growing literature on implicit regularization in RL appears somewhat incomplete. Could the authors clarify how this result compares to other recent analyses (e.g., Liu et al. (2024), Qiao et al. (2026), Xu et al. (2025)) in related work? Should PMD-MEAN be viewed as a distinct mechanism, or as another instance of the same broader phenomenon?

   [1] Huang, Audrey, et al. “Correcting the Mythos of KL-Regularization: Direct Alignment without Overoptimization via Chi-Squared Preference Optimization.” ICLR 2025.

   [2] Liu, Zhihan, et al. “Provably Mitigating Overoptimization in RLHF: Your SFT Loss is Implicitly an Adversarial Regularizer.” NeurIPS 2024.

   [3] Qiao, Nan, et al. “Less Is More: Clustered Cross-Covariance Control for Offline RL.” ICLR 2026.

   [4] Xu, Haoran, et al. “Uni-RL: Unifying Online and Offline RL via Implicit Value Regularization.” NeurIPS, 2025.

**Limitations:**

yes

**Strengths And Weaknesses:**

**Strengths**

- **[S1]** The core insight is genuinely interesting. Theorem 3.1 and Proposition 3.2 do more than justify a heuristic after the fact: they identify PMD-MEAN as solving a different regularized problem, with an adaptive KL–χ² structure rather than plain KL. That is a useful conceptual bridge between practical LLM RL training and regularized policy optimization theory.

- **[S2]** The analysis is not merely cosmetic. Section 4 articulates a concrete tradeoff: PMD-PART has faster ideal contraction, but PMD-MEAN is less sensitive to finite-sample target estimation error when rollouts are limited, especially early in training when reward mass on positive actions is small. Figures 3, 5, and 6 qualitatively support this story.

  ---
**Weaknesses**

- **[W1]** The biggest issue is the gap between the theory and the implemented objective. The exact characterization in Sections 3–4 is for sequence-level squared regression in log-policy space, but Appendix A implements a length-normalized loss of the form $\tau/|y|(\log \pi_\theta(y|x)/\pi_t(y|x)-A/\tau)^2$. That $1/|y|$ weighting is not innocuous. It changes the effective objective unless sequence lengths are constant or an invariance argument is provided. As written, the paper’s headline theorem does not precisely characterize the loss actually optimized in experiments. More broadly, the convergence analysis assumes contextual bandits, binary rewards, realizability, a finite policy class, and bounded log-ratios. This is useful intuition, but not a rigorous guarantee for neural autoregressive LLM training.
- **[W2]:** The empirical package is somewhat narrow for the scope of the claims. The main LLM evaluation uses a single training dataset (DAPO-Math-17k), two evaluation sets (AIME 2024/2025), and two base models. I also could not find multi-seed reporting or uncertainty intervals for Tables 1–3. The only explicit averaging over 100 random seeds appears in the stylized target-mismatch simulations in Figures 3 and 6, not in the main LLM results. The baseline picture is also incomplete for the paper’s central thesis: while comparisons to GRPO/GSPO are useful, the paper does not empirically compare against stronger regression-based PMD-style alternatives discussed in Related Work, such as Richemond et al. (2024) or Gao et al. (2024). Finally, the efficiency result in Table 2 is better interpreted as a throughput advantage of the stale, larger-global-batch training regime enabled by PMD-MEAN, since the paper itself notes that actor-update cost is comparable and attributes the 4.6× gain to amortized inference cost, rather than to a pure objective-level speed advantage.
- **[W3]**  The paper is generally readable, but Section 4 is dense and the bridge from the population analysis to the actual training recipe is not clear enough. To reduce cognitive load and improve narrative flow, I strongly recommend splitting Section 4 into two distinct chapters. Specifically, keep the general Inexact-PMD convergence framework as Section 4, and elevate the specific binary reward analysis (currently Section 4.2) into a new Section 5. This will much better highlight the core theoretical separation and stability advantages of PMD-MEAN under finite rollouts.
- **[W4]**  Some claims are also overstated relative to the evidence: the abstract says “superior performance,” yet Table 3 shows PMD-MEAN is only comparable to GSPO on the 30B model, not better.
- **[W5]** The proof for the existence and uniqueness of the normalization constant $\lambda$ in Appendix B.1 contains a minor logical leap. The authors argue that $\lambda$ is unique because the RHS of Eq. (27) is monotonically decreasing w.r.t. $\lambda$ and $\mathbb{E}[u] \le 0$ (via Jensen's inequality). However, $\mathbb{E}[u] \le 0$ is merely a necessary condition for a valid probability distribution and does not strictly prove the existence of a root for the normalization constraint.

Nevertheless, I believe this work is valuable, and I would like to discuss more.

---

> ### Author Rebuttal · Authors · 2026-03-31
>
> We thank the reviewer for the positive rating and valuable comments, and we respond to the concerns in the following.
>
> > **[W1&Q1] Theory-Implementation Gap.**
>
> Thanks for raising this point. We agree that the paper should be more explicit about the distinction between the exact sequence-level analysis and the loss in Appendix A. Theorem 3.1 and Proposition 3.2, as stated, characterize the unnormalized sequence-level regression objective, whereas Appendix A implements the length-normalized loss. So the theorem does not literally apply as written to the experimental objective, and we will state this explicitly. The same KKT derivation can, however, be extended to the normalized loss, yielding a Lambert-W characterization of a closely related mirror descent subproblem with length-dependent weighting (a length-weighted analogue of the mixed KL–$\chi^2$ regularization). We will add the corresponding remark and derivation in the revision. More broadly, Section 4 is intended as a stylized mechanism analysis under contextual bandit, realizability, and bounded ratio assumptions, not as a full theorem for LLM RL post-training.
>
> ---
>
> > **[W2&W4] Narrow Empirical Package and Overstated Claims Relative to Evidence**.
>
> We agree that the current empirical package is targeted rather than exhaustive, and that the main LLM results are single-run due to compute constraints. We will state this limitation more explicitly. We also agree that Table 2 should be interpreted as a throughput advantage from stale, larger-rollout-batch training rather than an objective-level speed advantage, and that "superior performance" should be revised to "improved stability and competitive performance", with the experimental section framed more clearly as mechanistic validation rather than a universal benchmark claim.
>
> To strengthen the empirical evidence, we expand the eval set to AIME26 and AMC23 and run additional Qwen3-8B-Base experiments, reporting the avg@32 in the following:
>
> | Staleness | Method | Avg | AIME24 | AIME25 | AIME26 | AMC23 |
> |---:|---|---:|---:|---:|---:|---:|
> | 16 | PMD-mean (tau=0.01)  | 43.0 | 34.9 | 29.2 | 26.9 | 81.0 |
> |    | GSPO | 42.1 | 32.7 | 29.4 | 28.3 | 77.9 |
> |    | GRPO | 35.8 | 26.8 | 22.2 | 23.2 | 70.9 |
> | 64 | PMD-mean (tau=0.01)  | 42.0 | 34.5 | 28.3 | 26.0 | 79.2 |
> |    | GSPO | 37.1 | 27.7 | 23.3 | 23.3 | 74.1 |
> |    | GRPO | 36.9 | 27.1 | 24.2 | 24.3 | 72.1 |
>
> These additional results support the same qualitative conclusion: PMD-mean remains competitive and degrades gracefully under stale rollouts. We will provide full details of these supplementary experiments in the revision.
>
> We also agree that broader comparisons to other regression-based PMD-style methods would strengthen the paper. We did not include Richemond et al. and Gao et al. because they require materially different training setups from our current critic-free, arbitrary group size RLVR pipeline (e.g., an additional value/critic model or pairwise relative reward fitting), so our empirical comparisons focused on methods used in the same practical regime. We will clarify this omission more explicitly as a limitation.
>
> ---
>
> > **[W3] Dense Presentation**.
>
> Thanks for your valuable suggestion. We agree Section 4 is dense and will separate the general inexact-PMD framework from the binary-reward instantiation to improve readability.
>
> ---
>
> > **[W5] Proof Gap (App. B.1)**.
>
>
> Thanks for catching this. You are right that there is a leap in the argument. A clean fix is to define $F(\lambda)=\sum_y\pi_t(y)e^{u_\lambda(y)}$, where $u_\lambda(y)=\frac{\Delta_y}{\tau}-W(\frac{\lambda}{\tau^2}e^{\Delta_y/\tau})$. Then $F(\lambda)$ is continuous and strictly decreasing on $[0,\infty)$ given strict monotonicity of $W(\cdot)$ on $[0,\infty)$ and that $\frac{e^{\Delta_y/\tau}}{\tau^2}>0$. Moreover, $F(0)=\mathbb E_{\pi_t}[e^{\Delta_y/\tau}]\ge 1$ by Jensen's inequality, and $F(\lambda)\to 0$ as $\lambda\to\infty$ because $W(z)=O(\log z)$. Therefore there exists a unique $\lambda^\star\geq 0$ with $F(\lambda^\star)=1$ by the intermediate value theorem, concluding the proof.
>
> ---
>
> > **[Q2] Implicit Regularization Context:** How does PMD-MEAN relate to recent implicit regularization works (e.g., Liu 2024 [2], Qiao 2026 [3], Xu 2025 [4])? Is it a distinct mechanism or part of a broader phenomenon?
>
> We agree that the paper should better situate itself within the larger literature on implicit regularization in RL. Our view is that PMD-mean belongs to this broader theme, but the mechanism here is distinct. The regularizer emerges at the mirror descent level through log-partition approximation in an online PMD regression objective, yielding an adaptive mixed KL–$\chi^2$ regularization. This differs from implicit regularization via preference+SFT coupling in [2], via TD-covariance control in [3], or via implicit value regularization around a reference policy in [4]. We will expand the related work discussion accordingly.

---

> > ### Author Rebuttal · Reviewer_kzUB · 2026-04-01
> >
> > The rebuttal addresses most concerns constructively and clarifies several points, but the central issues, especially the theory-implementation mismatch and the limited empirical validation, are only partially resolved at the rebuttal stage. Therefore, I keep my score unchanged.

---

### Official Review · Reviewer_8akB · 2026-03-19

**Soundness:** 3
**Presentation:** 4
**Significance:** 2
**Originality:** 2
**Overall Recommendation:** 4
**Confidence:** 3

**Summary:**

The paper investigates PMD-MEAN, a variant of Policy Mirror Descent (PMD) recently utilized in the post-training of large language models like Kimi K1.5 and K2. The authors prove that the population solution of PMD-MEAN is characterized by the Lambert-W function. Mathematically, this is equivalent to solving a mirror descent subproblem with an adaptive mixed $KL-\chi^2$ regularizer. The implicit $\chi^2$ regularization heavily penalizes large probability changes, making the updates significantly more conservative and robust to finite-sample estimation errors, especially when expected rewards are low during early training. Empirical evaluations on math reasoning tasks using Qwen 7B and 30B MoE models show that PMD-MEAN provides better stability and superior performance compared to baselines like GRPO and GSPO.

**Compliance With Llm Reviewing Policy:**

Affirmed.

**Final Justification:**

Weak accept: This is mainly a theory paper (little experiments) and the theory is novel and interesting but rely heavily on many unrealistic assummptions, such as the assumption of binary rewards. While this is standard for math and logic tasks, it somewhat limits the immediate generalizability of the theoretical claims to continuous or dense reward structures.

**Key Questions For Authors:**

Reward Structures: Much of the theoretical separation regarding ideal convergence rates and log-ratio bounds (Section 4.2) assumes a binary reward model. How does the implicit $\chi^2$ regularization behave mathematically under continuous or multi-level reward formulations?

Large Sample Limits: In Remark 4.12, you note that the gap between the ideal target and the estimated target of PMD-MEAN does not vanish as $n \to \infty$, but the minimizer of the empirical loss still recovers the ideal target policy due to the expectation constraint. Could you provide further intuition or preliminary analysis on how the optimization dynamics resolve this non-vanishing gap in practice?

Staleness Sensitivity: You demonstrate that PMD-MEAN handles a staleness of 16 effectively. Given that the algorithm naturally mitigates the "staleness tax", how sensitive is it to extreme off-policyness, can you expand the experimental section?

**Limitations:**

yes

**Strengths And Weaknesses:**

Originality:
Strengths: While the PMD-MEAN algorithm was introduced empirically by the Kimi team , this paper provides the first formal theoretical framework explaining its mechanics. Framing the mean-reward approximation as an implicit, adaptive mixed $KL-\chi^2$ regularizer (and deriving the closed-form Lambert-W solution) is a highly original and creative theoretical insight.

Weaknesses: The sharpest theoretical separations between PMD-MEAN and PMD-PART (Section 4.2) rely heavily on the assumption of binary rewards. While this is standard for math and logic tasks, it somewhat limits the immediate generalizability of the theoretical claims to continuous or dense reward structures.

Soundness:
Strengths: The theoretical derivations moving from the KKT conditions to the Lambert-W closed form are rigorous and well-supported. The convergence analysis effectively formalizes the intuition behind why PMD-PART fails and PMD-MEAN succeeds under finite rollouts. The experiments are appropriately scaled, utilizing modern architectures like a 30B MoE model to prove practical viability over state-of-the-art baselines like GRPO and GSPO.


Presentation:
Strengths: The paper is exceptionally well-structured. It clearly defines the gap between the ideal PMD update and the practical PMD-MEAN implementation early on. The narrative flows logically from establishing the population solution to analyzing convergence implications and finally providing empirical validation.

Significance:
Strengths: This paper addresses a highly relevant problem. RL post-training for LLMs is heavily reliant on algorithms that require complex, heuristic-heavy off-policy corrections (like sequence-level importance sampling with clipping) to maintain stability. By mathematically proving that PMD-MEAN inherently stabilizes training without these heuristics, this paper provides a robust theoretical foundation for practitioners to adopt simpler, more scalable RL algorithms.

Weakness: This impact is limited because no new regularization strategies or optimization strategies were attempted. The experimental section is already rather weak, with limited experiments and settings.

---

> ### Author Rebuttal · Authors · 2026-03-31
>
> We thank the reviewer for the positive feedback, and respond to the weaknesses and questions in the following.
>
> > **[W1&Q1] Binary Reward Assumption:** The sharpest theoretical separations (Sec 4.2) rely heavily on binary rewards, limiting generalizability to continuous/dense rewards. How does the implicit $\chi^2$ regularization behave mathematically under continuous or multi-level reward formulations?
>
> Thanks for this good question. For general rewards, the tight upper bounds on the ratio $\frac{\pi\_{t+1}}{\pi\_t}$ in Eq. (8) and (9) do not hold. Instead, we have upper bounds in Eq. (13) and (14), which read $O(\frac{1}{\tau})$ for the implicit $\chi^2$ regularization, and $O(\exp(\frac{1}{\tau}))$ for the standard KL regularization, implying a **polynomial vs. exponential** separation between the two regularizers. In this case, the $B_+$ in Assumption 4.3 is much smaller for the mixed $\chi^2$ regularization, leading to a better bound in Eq. (22).
>
> Meanwhile, **Theorem 3.1 and Proposition 3.2 do not require binary rewards**. Therefore, the mechanism of implicit regularization and the Lambert-W characterization of the optimal solution both hold for general rewards.
>
>
> ---
>
> > **[W2] Limited Empirical Impact:** The experimental section is weak because no new regularization or optimization strategies were attempted.
>
> We agree that this paper is primarily explanatory rather than proposing a new regularization or optimization strategy. Our main contribution is to characterize the objective implicitly optimized by PMD-mean and explain why it is more stable than PMD-part under stale rollouts. Accordingly, the experiments are intended as mechanistic validation of this theory rather than as a broad benchmark claim. While PMD-mean has been mentioned in prior reports [1], those works do not provide this characterization or an isolated controlled study of the approximation in our setting. We will make this positioning more explicit in the revision and calibrate the empirical claims accordingly.
>
> ---
>
> > **[Q2] Optimization Dynamics:** Regarding Remark 4.12, how do the optimization dynamics resolve the non-vanishing gap ($n \to \infty$) between the ideal and estimated target in practice?
>
> Thank you for raising this important point. We agree that the large-sample behavior should be stated more clearly.
>
> The non-vanishing gap mentioned in Remark 4.12 is not a fundamental inconsistency of PMD-mean. Instead, it comes from the coarse decomposition used in Lemma 4.5, which compares the Lambert-W population log-ratio target with the empirical advantage target pointwise. The latter is not an unbiased estimator of the former, since the PMD-mean population target includes the normalization induced by the simplex constraint.
>
> However, this pointwise mismatch does not imply an asymptotic error floor at the policy level, and we expect a refined loss-based analysis to remove this proof artifact. We will clarify this intuition more carefully in the revision.
>
>
> ---
>
> > **[Q3] Staleness Sensitivity:** How sensitive is the algorithm to extreme off-policyness beyond a staleness of 16? Can experiments be expanded to show this?
>
> Thanks for raising this question. To stress test extreme off-policyness, we conduct additional experiments with Qwen3-8B-Base, expanding the staleness to 64 and evaluation sets to AIME26 and AMC23, and report the avg@32 in the following.
>
> | Staleness | Method | Avg | AIME24 | AIME25 | AIME26 | AMC23 |
> |---:|---|---:|---:|---:|---:|---:|
> | 16 | PMD-mean (tau=0.01)  | 43.0 | 34.9 | 29.2 | 26.9 | 81.0 |
> | 16 | GSPO | 42.1 | 32.7 | 29.4 | 28.3 | 77.9 |
> | 16 | GRPO | 35.8 | 26.8 | 22.2 | 23.2 | 70.9 |
> | 64 | PMD-mean (tau=0.01)  | 42.0 | 34.5 | 28.3 | 26.0 | 79.2 |
> | 64 | GSPO | 37.1 | 27.7 | 23.3 | 23.3 | 74.1 |
> | 64 | GRPO | 36.9 | 27.1 | 24.2 | 24.3 | 72.1 |
>
> In these stale-rollout settings, PMD-mean is competitive with or stronger than the baselines we tested. We therefore view the main takeaway as robustness of PMD-mean under very stale rollouts.
>
> [1] Team, Kimi, et al. "Kimi k1.5: Scaling reinforcement learning with llms." arXiv preprint arXiv:2501.12599 (2025).

---

> > ### Author Rebuttal · Reviewer_8akB · 2026-04-01
> >
> > For reasons above which the authors agreed, I would make this a weak accept as it is mainly theoretical contribution in a limited setting.

---

### Official Review · Reviewer_xRpm · 2026-03-20

**Soundness:** 3
**Presentation:** 3
**Significance:** 2
**Originality:** 2
**Overall Recommendation:** 4
**Confidence:** 4

**Summary:**

The work examines the Policy Mirror Descent variant considered in the Kimi K1.5/K2 works. In regular PMD, the target policy is roughly pi_new \propto pi_old * exp(r/tau)  (which is treated as a regression target), and the difficult part is finding the normalization constant. The Kimi-style methods estimate the log normalization constant as the mean return (as opposed to a proper log-mean-exp). They call this method PMD-Mean. The main aim of the current work is to explain what this PMD-Mean is actually optimizing. They show that the PMD-Mean update is essentially PMD, but not with a pure KL regularizer, but with a mixture of chi-squared and KL-regularization. They contrast this with another approach PMD-Part that tries to estimate the log partition by taking the empirical log-mean-exp (though note that this will be a biased estimate as the log is essentially moved inside the expectation, but it is asymptotically unbiased as the number of samples goes to infinity). The main conclusion in the comparison is that PMD-Mean is more conservative than PMD-Part in its updates (this can for example be seen in figure 2), and also the converged location can be substantially different (e.g., figure 1).

Experimentally, they test using Qwen 2.5-7B-Base and Qwen3-30B-A3B-Base models, and show some improvement of PMD-Mean compared to GRPO (the evaluation looks slightly better on math tasks, but training rewards seem identical), and also show that PMD-Part does not learn effectively.
They also compare against GSPO, and the performance is similar with the 30B model, but PMD-Mean seems slightly better on the 7B model.

The main contribution from my perspective is the comparison between PMD-Mean and PMD-Part. While PMD-Mean and PMD-Part are proposed/mentioned in the Kimi works, the current work provides an explanation of what PMD-Mean is doing, and shows why it may be preferred over PMD-Part.

**Compliance With Llm Reviewing Policy:**

Affirmed.

**Final Justification:**

I kept my original score of weak accept because my original main reason for not setting a higher score was the scope of the work, which is limited to analyzing a particular known LLM post-training algorithm, and comparing it with another sensible idea that does not work well in practice. I found this interesting, but limited in breadth. I judged other aspects of the work such as the soundness and presentation to be good. The rebuttal was also well done in my opinion and clarified some of my concerns regarding the experiments and the performance of the GRPO baseline. However, my original review did not put a large emphasis on this comparison, because I viewed it as non-crucial to the claims the work is making.

**Key Questions For Authors:**

In Theorem 3.1., you define the advantage compared to the exact mean, but in practice, you’re still using a sampled mean. Can you close this gap somehow, e.g., is your sampling based method unbiased for this target objective that you proposed?

**Limitations:**

yes

**Strengths And Weaknesses:**

Strengths:

- The paper is clear (**good clarity**).

- The main claim in the paper is the explanation of PMD-Mean and PMD-Part and the evidence for this seems sufficient to claim that PMD-Mean performs better empirically, and also to explain the differences in performance. (**good soundness**)


Weaknesses:

- The scope of the work is not that large as it is essentially focused on specific PMD-Mean and PMD-Part methods and explaining them. PMD-Part is also not a commonly used method in the literature, and essentially the current work explains why that is the case. This is interesting, but it will not change what is done in practice. (**lower significance**)

- The main method PMD-Mean is not new. (**lower novelty**)

- The main method PMD-Mean is not new, so the experimental comparison results to GRPO etc. were not my focus, but the improvements are not completely convincing because the AIME datasets are quite small (30 questions each), which can lead to significant evaluation variance (e.g. random differences like 3-6% on evaluation performance would not be strange), and it seems that only 1 training run was done as well. This is quite common do to the training cost, so it is not a major concern for me (primarily because I don’t find the comparisons with GRPO/GSPO that crucial for this paper, and also because PMD-Mean being better than PMD-Part does seem clear from the results). Also, actually there are results online that show stronger results for GRPO on some tasks, e.g., on (https://openreview.net/pdf/4b6ae1cb9996eaa2f9b6ad66015bba402a1d7608.pdf) they show 50.63 for GRPO training on the 30B model, whereas the current paper shows 36.56. I am not very convinced that the PMD-Mean method is universally better than GRPO or other similar methods. (**lower soundness**)

---

> ### Author Rebuttal · Authors · 2026-03-31
>
> We thank the reviewer for the positive feedback, and respond to the weaknesses and questions in the following.
>
> > **[W1] Scope & Novelty:** The paper primarily explains existing methods (PMD-Mean/PMD-Part) rather than proposing a new algorithm.
>
> Thank you for the careful reading and for clearly identifying the main contribution of the paper. We agree with your framing that the contribution is not the introduction of PMD-mean as a new RL algorithm, but an exact characterization of the sequence-level PMD-MEAN objective and its population solution, together with an explanation of the stability mechanism relative to PMD-part in the finite-sample regime. We will make this more explicit in the abstract and introduction, and avoid wording that may suggest algorithmic novelty or universal empirical dominance.
>
> > **[W2] Limited Evaluation:** Improvements are not completely convincing due to small AIME datasets.
>
> Thanks for raising this concern. We acknowledge that AIME datasets are relatively small eval sets. To make the results more convincing, we conduct additional experiments with Qwen3-8B-Base, expanding the evaluation to AIME26 and AMC23, and report the avg@32 in the following.
>
> | Staleness | Method | Avg | AIME24 | AIME25 | AIME26 | AMC23 |
> |---:|---|---:|---:|---:|---:|---:|
> | 16 | PMD-mean (tau=0.01)  | 43.0 | 34.9 | 29.2 | 26.9 | 81.0 |
> |    | GSPO | 42.1 | 32.7 | 29.4 | 28.3 | 77.9 |
> |    | GRPO | 35.8 | 26.8 | 22.2 | 23.2 | 70.9 |
> | 64 | PMD-mean (tau=0.01)  | 42.0 | 34.5 | 28.3 | 26.0 | 79.2 |
> |    | GSPO | 37.1 | 27.7 | 23.3 | 23.3 | 74.1 |
> |    | GRPO | 36.9 | 27.1 | 24.2 | 24.3 | 72.1 |
>
> These additional results support the same qualitative conclusion: PMD-mean remains competitive and degrades gracefully under stale rollouts. We will provide full details of these supplementary experiments in the revision.
>
>
> > **[W3] GRPO Baseline:** Online results show stronger GRPO performance on the 30B model (e.g., 50.63 vs the reported 36.56). I am not very convinced that the PMD-Mean method is universally better than GRPO or other similar methods.
>
> Thanks for pointing this out. We agree that our paper should not be read as claiming universal superiority over GRPO or related methods. The 50.63 AIME24 result in [1] comes from a materially different, much less stale regime (1 ministep / staleness 1), whereas our main experiments use staleness 16, so the numbers are not directly comparable. [1] also reports degradation as staleness increases. Our empirical claim is narrower: PMD-mean is more stable and remains competitive in the high-staleness settings we study. We will revise the wording accordingly, replacing "superior performance" with more precise phrasing such as "improved stability and competitive performance in our evaluated stale-rollout settings."
>
>
> > **[Q1] Sampling Bias:** Theorem 3.1 compares against the exact mean, but practice uses a sampled mean. Is the sampling method unbiased for the proposed target objective? Can this gap be closed?
>
> The LOO mean is unbiased for the regression target $\Delta/\tau$ in Eq. (4), but not for the normalized Lambert-W population log-ratio target $s^\star$. This is why Proposition 4.11 contains a mismatch term. At the same time, this pointwise mismatch does not imply an asymptotic policy bias, as feasible log-ratios must satisfy $\mathbb{E}_{\pi\_t}[e^{s\_\pi}]=1$, thus the empirical minimization cannot simply fit the sampled advantages exactly. As discussed in Remark 4.12, the normalization constraint pulls the solution back towards the Lambert-W population policy in the large sample limit.
>
>
> [1] Ma, Wenhan, et al. "Stabilizing moe reinforcement learning by aligning training and inference routers." arXiv preprint arXiv:2510.11370 (2025).

---

> > ### Author Rebuttal · Reviewer_xRpm · 2026-04-04
> >
> > I appreciate the additional explanations and experiments, and the rebuttal is done well. My main reason for not setting a higher score was the scope of the work, where the authors agreed with my understanding of the scope. So I will maintain my score at the current weak accept, and remain positive about this work.

---

### Decision · Program_Chairs · 2026-04-30

**Decision:**

Accept (regular)

**Comment:**

This paper studies PMD-MEAN, an off-policy regression variant of KL-regularized policy mirror descent for LLM post-training, replacing the log-partition with mean rollout rewards and fitting log-policy ratios via squared loss. It shows that the population solution follows a Lambert-W update equivalent to mirror descent with a mixed KL–χ² regularizer. A stylized analysis suggests this yields more conservative updates and improved robustness to finite-sample errors.

Most reviewers agreed that the paper merits acceptance, and after carefully reviewing the rebuttal and discussion, I share this view. That said, I recommend that the authors incorporate the reviewers’ feedback into the final version of the paper, with particular attention to:

1- The theoretical analysis primarily assumes binary rewards; the paper should clarify why this assumption is not restrictive.

2- The MDP formulation used is uncommon in practice, which limits the scope and impact; a discussion on generalizability is needed.

3- The work does not explore alternative regularizers or optimization methods.